# GraphECL: Towards Efficient Contrastive Learning for Graphs

## Abstract

Due to the inherent label scarcity, learning useful representations on graphs with no supervision is of great benefit. Yet, existing graph self-supervised learning methods overlook the scalability challenge and fail to conduct fast inference of representations in latency-constrained applications due to the intensive message passing of graph neural networks. In this paper, we present `GraphECL`, a simple and efficient contrastive learning paradigm for graphs. To achieve inference acceleration, `GraphECL` does not rely on graph augmentations but introduces cross-model contrastive learning, where positive samples are obtained through `MLP` and `GNN` representations from the central node and its neighbors. We provide theoretical analysis on the design of this cross-model framework and discuss why our `MLP` can still capture structure information and enjoys better downstream performance as `GNN`. Extensive experiments on common real-world tasks verify the superior performance of `GraphECL` compared to state-of-the-art methods, highlighting its intriguing properties, including better inference efficiency and generalization to both homophilous and heterophilous graphs. On large-scale datasets such as Snap-patents, the `MLP` learned by `GraphECL` is 286.82x faster than GCL methods with the same number of GNN layers. Code and data are available at: https://github.com/GraphECL.

## 1 Introduction

Representation learning on graphs has been at the center of various real-world applications, such as drug discovery (Bongini et al., 2021), social analysis (Sankar et al., 2021) and anomaly detection (Zhao et al., 2021). In recent years, graph neural networks (GNNs) (Hamilton et al., 2017a; Kipf & Welling, 2017; Velickovic et al., 2018) have shown great power in node representation learning, and achieved remarkable performance on numerous node-related tasks. While learning representations on graphs is important, labeling graph data is challenging and resource intensive especially for those requiring domain knowledge, such as medicine and chemistry (Zitnik et al., 2018; Hu et al., 2020).

Fortunately, with the large amount of unlabeled graph data, self-supervised learning have shown promising results in representation learning without label information. Recently, graph contrastive learning (GCL), a popular self-supervised learning approach, is introduced to graph data due to the lack of task-specific labels (Veličković et al., 2018; Zhu et al., 2021; Thakoor et al., 2021; Zhang et al., 2021a). GCL has achieved competitive (or even better) generalization performance on many graph benchmarks compared to counterparts trained with ground-truth labels (Zhu et al., 2021; Thakoor et al., 2021).

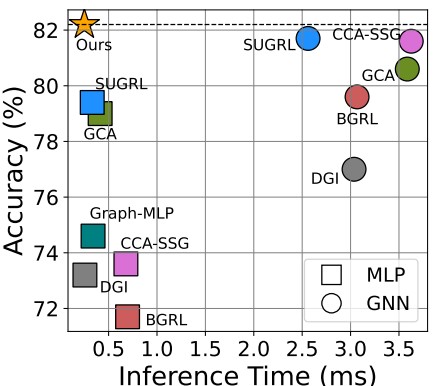

Figure 1: Inference latency v.s Accuracy. All methods are trained on Pubmed. For GCL baselines, we test them on both `MLP` and `GNN` backbones. We can observe that current SOTA methods require GNN as the encoder to achieve good performance, which is computation-intensive during inference. Our `GraphECL` is **20× faster** than SOTA methods with even **higher accuracy**.

Yet, it is challenging to scale GCL methods to large-scale applications which are constrained by latency and require fast inference, since the message passing of the `GNN` encoder involves fetching

topology and features from numerous neighboring nodes to perform inference on a target node, which is time-consuming and computation-intensive. In particular, we observe that utilizing a GNN encoder is the core recipe for existing GCL methods to achieve notable performance, as illustrated in Figure 1. As a result, the stronger performance of current GCL methods often comes at the cost of increased inference scalability. Nevertheless, less attention has been paid to designing GCL algorithms that have strong inference scalability, although it is important especially for the practical inductive setting where we need to conduct fast inference of representations for unseen test nodes.

To tackle the inference latency of GNN, recently, many efforts (Zheng et al., 2021; Zhang et al., 2021b; Tian et al., 2022; Wu et al., 2023) have been made on developing knowledge distillation (KD) (Hinton et al., 2015) to learn computationally-efficient student MLP by mimicking the logits outputted by the teacher GNN. However, these methods require task-specific labels to firstly train a good teacher GNN, making it infeasible to be adopted for GCL in practice. Prior works have (to be best of our knowledge) not addressed the scalability problem for GCL, which is crucial for real-time deployment of real-world graph applications. Consequently, the fundamental question that relates to the applicability of GCL remains unanswered, i.e., *How to design a graph contrastive learning algorithm such that strong inference scalability and generalization are simultaneously achieved?*

To answer this question, we present `GraphECL`, a *simple, effective, and efficient* contrastive regime for node representation learning. Specifically, to capture the neighborhood structure of nodes and maintain strong inference scalability, `GraphECL` introduces a cross-model contrastive architecture in which positive pairs consist of cross-model pairs (e.g., `MLP-GNN`) directly derived from neighborhood relations. These positive samples are obtained from the `MLP` and `GNN` representations of central nodes and their neighbors, respectively. This simple architecture allows `GraphECL` to benefit from graph context-awareness during training via `GNN` while having no graph dependency in inference through the `MLP`. Based on this cross-model architecture, we further propose a *generalized contrastive loss*, which facilitates the learning of a significantly faster `MLP` encoder, allowing it to effectively capture graph structural information and achieve impressive performance on downstream tasks.

To support practical evaluation, we benchmark `GraphECL` against state-of-the-art (SOTA) GCL schemes on various diverse 11 datasets. Rigorous experiments confirm the effectiveness and efficiency of `GraphECL` in learning structural information in graphs. Notably, `GraphECL` demonstrates strong generalization capabilities across diverse graph types, encompassing both homophilic and heterophilic graphs. `GraphECL` achieves significantly faster inference speeds than state-of-the-art GCL methods.

**Our major contributions are**: (i) We identify the limitation of current GCL methods and uncover that they are not effective or efficient. Thus, we study a novel problem of achieving satisfactory efficiency and accuracy simultaneously in GCL. (ii) We design `GraphECL`, a simple framework that can effectively learn graph structural information and conduct fast inference with a simple MLP. (iii) We theoretically characterize how `GraphECL` gradually captures structural information with `MLP` and prove that `GraphECL` can achieve provably generalization performance in downstream tasks. (iv) We demonstrate through extensive experiments that `GraphECL` can achieve ultra fast inference speed and superior performance at the same time. Our simple `GraphECL` has great potential to serve as a strong baseline and inspire followup works in designing efficient contrastive learning algorithms on graphs.

## 2 RELATED WORK

**Graph Contrastive Learning.** GCL has gained popularity in representation learning, which aims to learn useful representations from fully unlabeled graph data (Veličković et al., 2018; Zhu et al., 2021; You et al., 2020; Suresh et al., 2021; Zhang et al., 2021a). The basic idea is to maximize the similarity between views augmented from the same instances and optionally minimize the similarity between views augmented from different instances. Recently, advanced methods are proposed to free GCL from negative samples (Thakoor et al., 2021; Zhang et al., 2021a) or even graph augmentations (Xia et al., 2022a; Lee et al., 2022; Zhang et al., 2022). Some approaches accelerate GCL training (Mo et al., 2022; Zheng et al., 2022) to a certain extent. Despite this progress, current methods has high neighbor-fetching latency, i.e., the message passing to aggregate information from neighbors, limiting their scalability during inference (Zhang et al., 2021b). In contrast, we aim to enhance inference efficiency while maintaining the effectiveness of GCL. Our work enjoys the benefits of graph context-awareness but has no graph dependency during the inference process.

**Learning MLPs on Graphs.** Recent works (Yang et al., 2022a; Han et al., 2022) find that using MLPs in training and GNNs in inference can significantly accelerate training. They primarily focus on semi-supervised settings with task-specific labels and still require time-consuming neighbor retrieval during inference. Our work is also related to graph-regularized MLP (Yang et al., 2016; Hu et al., 2021; Yang et al., 2021b; Liu et al., 2020), which incorporates graph structure into MLPs through various auxiliary regularization terms inspired by traditional network embedding methods (Hamilton et al., 2017b; Grover & Leskovec, 2016; Tang et al., 2015). By implicitly encoding structural information into MLPs, one can enhance the representational power of `MLP` encoders while maintaining fast inference. However, a significant performance gap exists between these graph regularized terms and GCL in unsupervised settings (Veličković et al., 2018; You et al., 2020). Moreover, it's worth noting that these methods, despite their minor variations, are all based on the homophily assumption (McPherson et al., 2001), which posits that connected one-hop neighbors should exhibit similar latent representations. However, recent studies (Zhu et al., 2020a; Lim et al., 2021; Chien et al., 2020) have demonstrated that real-world heterophilic graphs violate this assumption, leading these methods to potentially struggle with generalization. In contrast to these past work, we aim to achieve both strong inference scalability and generalization on both homophilous and heterophilous graphs for the first time.

**Knowledge Distillation on Graphs.** Knowledge distillation on graphs, which aims to distill pre-trained teacher GNNs into smaller student GNNs, has recently garnered significant attention (Yang et al., 2020; 2022b; Yan et al., 2020; Yang et al., 2021a; Joshi et al., 2022). Since student GNNs still require time-consuming message passing in the inference, recent studies (Yang et al., 2023; Zhang et al., 2021b; Zheng et al., 2021; Tian et al., 2022; Wu et al., 2023) have shifted their focus towards GNN-MLP distillation. This involves learning a computationally-efficient student MLPs by distilling knowledge from teacher GNNs. However, these methods typically rely on task-specific labels to train the teacher GNN, which can be challenging in real-world scenarios where labels are often inaccessible. In contrast, our work focuses on graph contrastive learning, with the aim of developing a computationally-efficient and structure-aware MLP that does not require labels.

## 3 PRELIMINARIES

**Problem Setup.** The input graph is denoted as $G = (\mathcal{V}, \mathcal{E})$, where $\mathcal{V} = \{v_1, \ldots, v_{|\mathcal{V}|}\}$ is a set of $|\mathcal{V}|$ nodes and $\mathcal{E}$ denotes the set of edges. Each edge $e_{i,j} \in \mathcal{E}$ denotes a link between nodes $v_i$ and $v_j$. We use $\mathbf{X} \in \mathbb{R}^{|\mathcal{V}| \times D}$ to denote the node attribute matrix, where $i$-th row of $\mathbf{X}$, i.e., $\mathbf{x}_i$, is the attribute vector of node $v_i$. The graph structure can be characterized by its adjacency matrix $\mathbf{A} \in [0, 1]^{|\mathcal{V}| \times |\mathcal{V}|}$, where $\mathbf{A}_{i,j} = 1$ if there exists an edge $e_{i,j} \in \mathcal{E}$, and $\mathbf{A}_{i,j} = 0$ otherwise. Then, the graph $\mathcal{G}$ can be also denoted as a tuple of matrices: $G = (\mathbf{X}, \mathbf{A})$. Given $G$, our goal is to learn an efficient `MLP` encoder denoted by $f_M$ with only attributes $\mathbf{X}$ as the input, such that the inferred representation for node $v$: $\mathbf{v} = f_M(\mathbf{X})[v] \in \mathbb{R}^K$ is useful and generalized for various downstream tasks. For brevity, we omit the input $\mathbf{X}$ and use $f_M(v)$ to denote $v$'s representation from `MLP` in the paper.

**Graph Contrastive Learning (GCL) with Augmentations.** GCL aims to learn representations (Trivedi et al., 2022; Veličković et al., 2018; Zhu et al., 2021; You et al., 2020; Suresh et al., 2021) through the contrast of augmented views as presented in Figure 2 (a). Specifically, for a given node $v$, its representation in one augmented view is trained to be similar to the representation of the same node $v$ from another augmented view, while being distinct from the representations of other nodes serving as negative samples. Given two views $G_1$ and $G_2$, one widely-used contrastive objective is:

$$\mathcal{L}_{\text{GCL}} = -\frac{1}{|\mathcal{V}|} \sum_{v \in \mathcal{V}} \log \frac{\exp(f_G(v^1)^\top f_G(v^2)/\tau)}{\exp\left(f_G(v^1)^\top f_G(v^2)/\tau\right) + \sum_{v^- \in \mathcal{V}^-} \exp\left(f_G(v^1)^\top f_G(v^-)/\tau\right)}. \quad (1)$$

Here $f_G(v^1) = f_G(G_1)[v]$ and $f_G(v^2) = f_G(G_2)[v]$ are `GNN` representations of the same node $v$ from two views, where $f_G$ denote the `GNN` encoder. $\mathcal{V}_-$ is the set of negative samples from inter- or intra- augmented view (Zhu et al., 2020b). $\tau$ is the temperature hyper-parameter. Although GCL with augmentations has achieved remarkable success, we identify that methods developed under this framework predominantly rely on `GNN` encoder to capture structural invariances across varied augmented views of the graph. This reliance is further discussed in Section 4.1 and leads to a comparative reduction in inference speeds against `MLP`, as depicted in Figure 1.

**Graph-regularized MLP.** Graph-MLP (Yang et al., 2016; Hu et al., 2021; Yang et al., 2021b; Liu et al., 2020) proposes to bypass `GNN` neighbor fetching by learning a computationally-efficient MLP

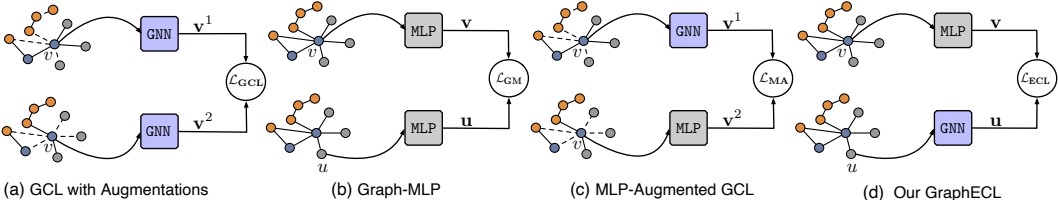

(a) GCL with Augmentations     (b) Graph-MLP     (c) MLP-Augmented GCL     (d) Our GraphECL

Figure 2: Simple illustration of existing contrastive schemes and `GraphECL`. (a) and (c) rely on invariant assumptions, aiming to learn augment-invariant representations of the same node. (b) is based on homophily assumptions, forcing neighboring nodes to exhibit same representations. In contrast, (d) showcases our `GraphECL`, which achieves inference efficiency using `MLP` and does not depend on invariant or homophily assumptions.

model with a neighbor contrastive loss inspired by traditional network embedding methods (Hamilton et al., 2017b; Grover & Leskovec, 2016; Tang et al., 2015). Despite the minor differences, they minimize the following unsupervised contrastive loss over neighbors in the graph:

$$\mathcal{L}_{\text{GM}} = -\frac{1}{|\mathcal{V}|} \sum_{v \in \mathcal{V}} \frac{1}{|\mathcal{N}(v)|} \sum_{u \in \mathcal{N}(v)} \log \frac{\exp(f_M(v)^\top f_M(u)/\tau)}{\exp\left(f_M(v)^\top f_M(u)/\tau\right) + \sum_{v^- \in \mathcal{V}^-} \exp\left(f_M(v)^\top f_M(v^-)/\tau\right)}, \quad (2)$$

where $f_M(v)$, $f_M(u)$ and $f_M(v^-)$ are projected representations by `MLP` of nodes $v$, $u$ and $v^-$, respectively. $\mathcal{N}(v)$ is the positive sample set containing local neighborhoods of central node $v$ and $\mathcal{V}^-$ is the set of negative sample which can be randomly sampled from $\mathcal{V}$. This paradigm is illustrated in Figure 2 (b). Despite its inference efficiency due to the exclusive use of `MLP`, this approach exhibits significantly lower performance compared to GCL with augmentations, as depicted in Figure 1. Moreover, this scheme over-emphasizes homophily, assuming that connected nodes should have similar representations in the latent space, at the expense of structural information (You et al., 2020; Xiao et al., 2022), making it difficult to generalize to graphs with heterophily (Lim et al., 2021).

Table 5 in Appendix A summarizes the detailed comparisons between current contrastive schemes on graphs and our `GraphECL` in terms of design assumptions, effectiveness and inference efficiency.

## 4 DESIGNING EFFICIENT CONTRASTIVE LEARNING FOR GRAPHS

In this section, we introduce `GraphECL`, a unified GCL framework that effectively addresses the efficiency limitations of inference while maintaining strong generalization capabilities. `GraphECL` adopts a cross-model contrastive architecture, wherein we design an asymmetric GNN-MLP architecture for central nodes and their neighbors to create effective positive and negative pairs in the context of contrastive learning (Section 4.1). In order to exploit the structural information in the graph, we propose *generalized contrastive loss* that extends the classic InfoNCE loss from *independent* instance discrimination over augmentations to *non-independent* neighborhood contrast over graph structures, which takes into account the meaningful distance between neighboring nodes. Finally, we provide a theoretical analysis to justify that `GraphECL` can still capture structural information and enjoys the better downstream performance with the simple `MLP` (Section 4.2).

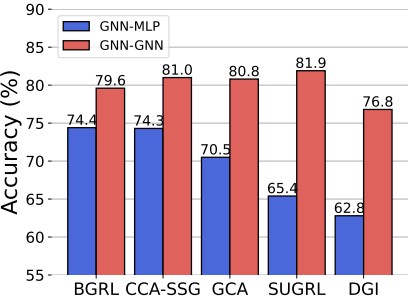

Figure 3: Current GCL methods that employ a `GNN-MLP` architecture (where `MLP` is used for inference) exhibit a significant performance decay compared to those using a `GNN-GNN` architecture (where `GNN` is used for inference). We illustrate Pubmed as an example, though we observe the similar trend in other datasets.

### 4.1 SIMPLE CROSS-MODEL CONTRASTIVE LEARNING FRAMEWORK

As motivated and shown in Figure 1, current state-of-the-art GCL methods suffer from inference latency due to the layer-wise message passing of `GNN` encoders. A straightforward idea is to instead utilize `MLP` as the encoder for those methods, allowing for fast inference. However, as seen in Figure 1, the performance of current GCL methods with `MLP` as the backbone is significantly worse than its `GNN` counterpart. Intuitively, these results align with our expectations that discarding message passing degrades them to a situation without using graph augmentations, making it impossible for them to

learn structural invariances in the augmented graph as originally designed. Thus, the architecture with encoder `GNN` is very important for GCL methods based on graph augmentations.

**Cross-Model Contrastive Architecture**. To address this limitation, we first introduce a simple cross-model architecture within `GraphECL`. As shown in Figure 1, using `MLP` as the encoder for GCL is computationally efficient during inference but yields suboptimal performance. Conversely, utilizing `GNN` yields superior results but necessitates time-consuming neighbor retrieval during inference. Our idea for this challenge is elegantly simple, yet, as we will demonstrate, remarkably effective. Specifically, we employ a cross-model architecture, wherein we utilize two encoders: one being a `GNN`, and the other an `MLP`. In contrast to past work (Mo et al., 2022; Zhu et al., 2021), `GNN` in this architecture is exclusively engaged in the learning process to capture structural information, while `MLP` is employed during the inference process to eliminate graph dependencies. One straightforward question is whether we can directly apply this architecture to current GCL methods. For instance, we can minimize the following loss, referred to as MLP-Augmented GCL as shown in Figure 2 (c):

$$\mathcal{L}_{\text{MA}} = -\frac{1}{|\mathcal{V}|} \sum_{v \in \mathcal{V}} \log \frac{\exp(f_G(v^1)^\top f_M(v^2)/\tau)}{\exp\left(f_G(v^1)^\top f_M(v^2)/\tau\right) + \sum_{v^- \in \mathcal{V}^-} \exp\left(f_G(v^1)^\top f_M(v^-)/\tau\right)}. \tag{3}$$

Unfortunately, the answer is negative. As depicted in Figure 3, even with cross-model architecture, current GCL methods suffer from a remarkably performance decay compared to exclusively utilizing `GNN` encoders in Equation (3). These results show that `MLP` with the invariant assumption of graph augmentations struggles to capture useful structural information with this cross-model architecture.

**Generalized Contrastive Learning Loss**. Therefore, we are naturally inspired to propose a new contrastive learning objective function. Figure 2 highlights the properties and differences between other paradigms and our `GraphECL`. Specifically, existing GCL methods adopt graph augmentations that emphasize strict "invariances" of the same node in two augmented views, and highly rely on `GNN` as the encoder. In contrast, `GraphECL` does not rely on any graph augmentations, but learns `MLP` representation of node by capturing its one-hop neighborhood signal from `GNN`. In particular, positive pairs in GCL are generated by random graph augmentations of the same node. In contrast, positive pairs in `GraphECL` are cross-model pairs (e.g., `MLP`-`GNN`) directly provided by neighborhood relations. These positive samples are obtained through `MLP` and `GNN` representations from central nodes and their neighbors, respectively. `GraphECL` aims to push the `MLP` representation of the central node closer to the `GNN` representations of its neighbors by minimizing the following loss:

$$\mathcal{L}_{\text{pos}} = \frac{1}{|\mathcal{V}|} \sum_{v \in \mathcal{V}} \frac{1}{|\mathcal{N}(v)|} \sum_{u \in \mathcal{N}(v)} \left\| f_M(v) - f_G(u) \right\|_2^2, \tag{4}$$

where $f_M(v)$ and $f_G(u)$ are the L2-normalized representations obtained from the `MLP` and `GNN` encoders, respectively, of node $v$ and its neighbor $u$. We highlight the benefits of using this simple alignment loss. First, the `MLP` encoder $f_M$ can effectively preserve the local neighborhood distribution captured by `GNN` encoder $f_G$ without requiring graph augmentation. Additionally, as demonstrated in Section 5, when `MLP` $f_M$ is used for downstream tasks, `GraphECL` continues to perform exceptionally well, offering fast inference and significantly outperforming GCL counterparts with `GNN`. Second, in contrast to the GR-MLP model depicted in Figure 2 (b), GraphECL aims to push cross-modal neighboring node representations closed, i.e., $(f_M(v), f_G(u))$. Thus, it's crucial to emphasize that GraphECL doesn't necessarily imply the learned MLP representations $(f_M(v), f_M(u))$ become identical. Consider a pair of 2-hop neighbors, $v$ and $v'$, both neighboring the same node $u$. Intuitively, by enforcing $f_M(v)$ and $f_M(v')$ to reconstruct (not align) the context representation of the same neighborhood $f_G(u)$, we implicitly make their representations similar. Thus, the 2-hop neighbors $(f_M(v)$ and $f_M(v'))$ with the same neighborhood context serve as positive pairs that will be implicitly aligned. This alignment is supported by our Theorem 4. Additionally, as depicted in Figure 9, GraphECL is not based on the one-hop homophily assumption but automatically captures graph structures based on different graphs beyond homophily. Consequently, GraphECL exhibits robustness and generalizability, effectively accommodating both homophilic and heterophilic graphs (Section 5).

While the alignment loss in Equation (4) effectively captures structural information, it is widely recognized that in contrastive learning, merely aligning positive pairs can lead to dimensional collapse (Hua et al., 2021; Jing et al., 2022), limiting their representational capacity. To address this problem, we introduce cross-model negative pairs, resulting the final objective of `GraphECL`:

$$\mathcal{L}_{\text{ECL}} = -\frac{1}{|\mathcal{V}|} \sum_{v \in \mathcal{V}} \frac{1}{|\mathcal{N}(v)|} \sum_{u \in \mathcal{N}(v)} \log \frac{\exp(f_M(v)^\top f_G(u)/\tau)}{\sum_{v^- \in \mathcal{V}} \exp(f_G(u)^\top f_G(v^-)/\tau) + \lambda \exp(f_M(v)^\top f_G(v^-)/\tau)}, \tag{5}$$

where the numerator term has the same effect as Equation (4) since the representations are normalized. Here, $(f_G(u), f_G(v^-))$ and $(f_M(v), f_G(v^-))$ represent intra-model and inter-model negative pairs, respectively. $v^-$ is independently sampled as a negative example, and $\lambda$ serves as a hyperparameter to control the balance between the two types of negative pairs. For large graphs, we employ random sampling to select $M$ negative pairs for each node as an efficient approximation during training.

**Interpretation.** $\mathcal{L}_{\text{ECL}}$ is a simple yet very effective generalization of the popular InfoNCE loss in Equation (3) from uni-model instance discrimination over augmentations to cross-model contrast over graph neighbors. During the learning process, cross-model positive pairs of neighbors $(f_M(v), f_G(u))$ are pulled together in the latent space, while inter-model $(f_M(v), f_G(v^-))$ and intra-model negative pairs $(f_G(u), f_G(v^-))$ are pushed apart. We theoretically prove why simple `MLP` in `GraphECL` can still capture structural information and enjoys the good downstream performance (Section 4.2). We also empirically demonstrate in Section 5 that `GraphECL` generalizes as well as state-of-the-art GCL methods during the inference, with the additional benefit of faster inference and easier deployment.

## 4.2 THEORETICAL ANALYSIS

In this section, we provide theoretical evidence to support the design of our simple `GraphECL`. All detailed proofs can be found in Appendix B. We start by providing some notations. We denote the normalized adjacency matrix $\mathbf{D}^{-1}\mathbf{A}$, with $\mathbf{D}$ being the diagonal degree matrix. We define the two representation metrics $\mathbf{M}$ and $\mathbf{G}$ where the $v$-th row $(\mathbf{M})_v = f_M(v)$ and the $u$-th row $(\mathbf{G})_u = f_G(u)$ represent the corresponding encoded representations from `MLP` and `GNN`, respectively. Let $\bar{\mathbf{A}} = \exp(\mathbf{M}\mathbf{G}^\top/\tau)$ is the estimated affinity matrix based on representation similarity. $\bar{\mathbf{D}} = \deg(\bar{\mathbf{A}})$ is the diagonal matrix, whose element $(\bar{\mathbf{A}})_{i,i}$ is the sum of the $i$-th row of $\bar{\mathbf{A}}$. Next, we reveal the stationary point of the learning dynamics of `GraphECL`, which implies the model equilibrium as:

**Theorem 4.1.** *The learning dynamics w.r.t the `MLP` encoder $f_M$ with the generalized contrastive loss ($\lambda = 1$) in Equation (5) saturates when the true normalized adjacency and the estimated normalized affinity matrices agree: $\mathbf{D}^{-1}\mathbf{A} = \bar{\mathbf{D}}^{-1}\bar{\mathbf{A}}$, which implies that, for $\forall v, u \in \mathcal{V}$, we have:*

$$\mathcal{P}_n(u \mid v) = \mathcal{P}_f(u \mid v) \triangleq \frac{\exp(f_M(v)^\top f_G(u)/\tau)}{\sum_{v' \in \mathcal{V}} \exp(f_M(v)^\top f_G(v')/\tau)}, \tag{6}$$

*where $\mathcal{P}_n(u \mid v)$ is the 1-hop neighborhood distribution (i.e., the $v$-th row of the normalized adjacency matrix) and $\mathcal{P}_f(u \mid v)$ is the Boltzmann distribution estimated by encoders with temperature $\tau$.*

Theorem 4.1 implies that `GraphECL` essentially learns a probabilistic model based on cross-model encoders to predict the conditional 1-hop neighborhood distribution. Specifically, our assumption is more general than the homophily assumption. Even in heterophilic graphs, two nodes of the same semantic class tend to share similar structural roles, i.e., 1-hop neighborhood context as shown in (Ma et al., 2021; Chien et al., 2021) and graph statistics in Appendix C.2. This can explain the effectiveness of the `MLP` encoder $f_M$ in learning effective representations some heterophilic graphs.

We establish formal guarantees for the generalization of `GraphECL` on downstream tasks for learned `MLP` and `GNN` encoders. Without loss of generality, we use the linear probing task as an example. In this task, we train a linear classifier to predict class labels $y \in \mathcal{Y}$ based on the `MLP` representation $f_M$ using $g_{f,W}(v) = \arg\max_{c \in [C]}(f_M(v)^\top W)_c$, where $W \in \mathbb{R}^{K \times C}$ represents the weight matrix.

**Theorem 4.2.** *Let $f_M^*$ be the global minimum of generalized contrastive loss ($\lambda = 1$) in Equation (5) and $y(v)$ denote the label of $v$. $\sigma_1 \geq \cdots \geq \sigma_N$ are the eigenvalues with descending order of the normalized adjacency matrix $\mathbf{D}^{-1}\mathbf{A}$. Then, the linear probing error of $f_M^*$ is upper-bounded by:*

$$\mathcal{E}(f_M^*) \triangleq \min_W \frac{1}{|\mathcal{V}|} \sum_{v \in \mathcal{V}} \mathbb{1}[g_{f^*,W}(v) \neq y(v)] \leq \frac{1 - \alpha}{1 - \sigma_{K+1}}, \tag{7}$$

*where $\alpha = \frac{1}{|\mathcal{V}|} \sum_{v \in \mathcal{V}} \frac{1}{|\mathcal{N}(v)|} \sum_{u \in \mathcal{N}(v)} \mathbb{1}[y(v) = y(u)]$ and $K$ is the dimension of the representation.*

This theorem establishes a significant relationship between the downstream error in learned representations and two crucial factors: the parameter $\alpha$ and the $(K+1)$-th largest eigenvalue. Remarkably, $\alpha$ coincides precisely with the node homophily ratio metric, as defined in prior works (Pei et al., 2019; Lim et al., 2021). This metric calculates the proportion of a node's neighbors that share the same class label and then averages these values across all nodes within the graph. Homophilous graphs

Table 1: Node classification results (%) under the transductive setting on benchmarking homophilic and heterophilic graphs. The best and runner up methods are marked with boldface and underline, respectively.

| Datasets | Cora | Citeseer | Pubmed | Photo | Flickr | Cornell | Wisconsin | Texas | Crocodile | Actor | Snap-patents |
|---|---|---|---|---|---|---|---|---|---|---|---|
| Graph-MLP | 76.70±0.18 | 70.30±0.27 | 78.70±0.33 | 89.59±0.45 | 41.33±0.25 | 42.65±2.21 | 57.96±1.11 | 60.22±1.76 | 53.22±2.31 | 25.66±0.77 | 21.41±0.62 |
| VGAE | 76.30±0.21 | 66.80±0.23 | 75.80±0.40 | 91.50±0.20 | 40.71±0.22 | 48.73±4.19 | 55.67±1.37 | 50.27±2.21 | 45.72±1.53 | 26.99±1.56 | 21.43±0.55 |
| DGI | 82.30±0.60 | 71.80±0.70 | 76.80±0.60 | 91.61±0.22 | 44.70±0.26 | 45.33±6.11 | 55.21±1.02 | 58.53±2.98 | 51.25±0.51 | 28.30±0.76 | 22.98±0.37 |
| GCA | 82.93±0.42 | 72.19±0.31 | 80.79±0.45 | 91.70±0.10 | 46.10±0.19 | 52.31±1.09 | 59.55±0.81 | 52.92±0.46 | 60.73±0.28 | 28.77±0.29 | 23.11±0.57 |
| SUGRL | 83.40±0.50 | 73.00±0.40 | 81.90±0.30 | 93.07±0.15 | 46.22±0.31 | 50.18±0.30 | 61.31±2.07 | 57.88±2.21 | 55.52±0.75 | 30.31±0.82 | 25.11±0.32 |
| BGRL | 82.70±0.60 | 71.10±0.80 | 79.60±0.50 | 92.90±0.30 | 45.33±0.19 | 50.33±2.29 | 51.23±1.17 | 52.77±1.98 | 53.87±0.65 | 28.80±0.54 | 24.33±0.13 |
| CCA-SSG | 84.00±0.40 | 73.10±0.30 | 81.00±0.40 | 93.14±0.14 | 47.54±0.14 | 52.17±1.04 | 58.46±0.96 | 59.89±0.78 | 56.77±0.39 | 27.82±0.60 | 25.51±0.46 |
| DSSL | 83.20±0.42 | 72.31±0.51 | 81.25±0.31 | 93.10±0.32 | 46.78±0.22 | 53.15±1.28 | 62.26±0.55 | 62.11±1.53 | 62.98±0.51 | 28.15±0.31 | 25.55±0.41 |
| AF-GCL | 83.16±0.13 | 71.96±0.42 | 79.16±0.73 | 92.49±0.31 | 46.95±0.33 | 52.29±1.21 | 60.12±0.39 | 59.81±1.33 | 61.72±0.21 | 28.94±0.69 | 26.31±0.75 |
| AFGRL | 81.30±0.20 | 68.70±0.30 | 80.60±0.40 | 93.22±0.28 | 46.81±0.20 | 55.37±3.56 | 63.21±1.55 | 60.35±1.05 | 60.31±0.87 | 30.31±0.95 | 24.26±0.81 |
| GraphECL | 84.25±0.05 | 73.15±0.41 | 82.21±0.05 | 94.22±0.11 | 48.49±0.15 | 69.19±6.86 | 79.41±2.19 | 75.95±5.33 | 65.84±0.71 | 35.80±0.89 | 27.22±0.06 |

Table 2: Node classification results in a real-world scenario with both inductive and transductive nodes. **tran** denotes the results on seen transductive nodes. **ind** indicates the accuracy on unseen inductive nodes.

| Methods | Citeseer | | Pubmed | | Photo | | Actor | | Flickr | |
|---|---|---|---|---|---|---|---|---|---|---|
| | tran | ind | tran | ind | tran | ind | tran | ind | tran | ind |
| DGI | 63.82±1.69 | 66.25±2.54 | 70.33±2.61 | 70.48±2.41 | 87.11±1.65 | 88.14±0.45 | 28.07±2.19 | 28.08±1.96 | 37.84±0.22 | 39.71±0.30 |
| GCA | 66.33±1.16 | 69.02±2.08 | 81.16±0.80 | 81.52±0.56 | 90.54±0.54 | 90.59±0.51 | 27.94±1.62 | 27.72±1.51 | 41.25±0.33 | 42.95±0.18 |
| BGRL | 67.04±1.44 | 67.62±1.24 | 78.36±0.41 | 79.55±0.40 | 87.95±0.68 | 88.30±0.45 | 29.04±1.06 | 29.07±0.65 | 40.78±0.20 | 41.75±0.15 |
| SUGRL | 69.16±0.63 | 71.24±1.06 | 81.07±0.76 | 80.52±1.21 | 89.88±0.64 | 89.11±0.24 | 28.95±1.37 | 28.68±1.18 | 40.37±0.20 | 41.33±0.25 |
| CCA-SSG | 68.81±1.05 | 70.05±2.70 | 79.76±2.32 | 80.34±2.32 | 88.60±1.95 | 88.77±1.85 | 28.52±1.11 | 28.06±2.69 | 42.16±0.25 | 43.22±0.27 |
| GraphECL | 69.96±0.10 | 72.87±1.30 | 81.71±0.91 | 82.47±1.00 | 92.18±0.15 | 89.42±0.03 | 36.18±1.29 | 37.17±1.84 | 45.43±0.14 | 43.50±0.20 |

($\alpha \to 1$), exhibit a tendency for nodes to connect with others of the same class, while heterophilic graphs ($\alpha \to 0$), display a preference for connections across different classes. This theorem shows that graphs characterized by a low homophily value (i.e., heterophilic graphs) may require a larger representation dimension, i.e., smaller ($K+1$)-th largest $\sigma_{K+1}$ to effectively bound the downstream error. We empirically verify the effect of dimensions on different real-world graphs (Section 5.3).

## 5 EXPERIMENTS

**Datasets.** We perform extensive experiments on datasets that span different domains. For homophilic graphs, we adopt the widely-used benchmarks: Cora, Citeseer, and Pubmed, co-purchase graph Photo. For heterophilic graphs, we utilize Cornell, Wisconsin, Texas, Crocodile and Actor. To fully evaluate our method, we also consider two large-scale real-world graphs: Flickr and Snap-patents. We adhere to the *public and standard splits* employed by the previous studies for all datasets. Detailed descriptions, splits, and statistics for these datasets can be found in C.3.

**Baselines.** We compare with recent graph contrastive or self-supervised methods: Graph-MLP (Hu et al., 2021), VGAE (Kipf & Welling, 2016b), DGI (Veličković et al., 2018), GCA (Zhu et al., 2021), SUGRL (Mo et al., 2022), BGRL (Thakoor et al., 2021), CCA-SSG (Zhang et al., 2021a), DSSL (Xiao et al., 2022), AF-GCL (Wang et al., 2022a), and AFGRL (Lee et al., 2022).

**Evaluation Protocol.** Following (Veličković et al., 2018; Thakoor et al., 2021; Zhu et al., 2021), we consider two downstream tasks: node classification and node clustering. For node classification, we use standard linear-evaluation protocol, where a linear classifier is trained on top of the frozen representation, and test accuracy is used as a proxy for representation quality. For node clustering, we conduct k-means clustering on the representations, setting the number of clusters equal to the number of ground truth classes, and report normalized mutual information (NMI) (Vinh et al., 2009).

**Transductive vs. Inductive.** The evaluation of unsupervised node representations through transductive node classification is a prevalent practice in GCL. Nevertheless, it neglects the scenarios of inferring representations for previously unseen nodes. Thus, it can not evaluate the real-world applicability of a deployed model, which often requires the inference of representations for novel data points. We consider evaluating representations under two settings: transductive (tran) and inductive (ind). The details about these two settings are given in Appendix C.4.

**Setup.** For a fair comparison, we employ a standard GCN model (Kipf & Welling, 2017) as the GNN encoder for all methods unless otherwise specified. We conduct experiments using ten random seeds and report both the average performance and standard deviation. We select the optimal hyperparameters solely based on accuracy on the validation set. In cases where publicly available and standardized data splits were used in the original paper, we adopt their reported results. For baselines

that deviated from publicly available and standardized data splits, we either reproduce the results using the authors' official code. The hyperparameter search space can be found in Appendix C.5.

## 5.1 MAIN RESULTS AND COMPARISON ON BOTH TRANSDUCTIVE AND INDUCTIVE SETTINGS

In this section, we evaluate the node representations from the `MLP` encoder learned by our `GraphECL`.

**Transductive Setting.** We first consider the standard transductive setting on the task of node classification. We provide the results of the node clustering task in Table 7 in Appendix. Table 12 reports the average accuracy on both heterophilic and homophilic graphs. As shown in the table, across different datasets, `GraphECL` is able to learn better representations that achieve the best performance. These results are profoundly encouraging and indeed remarkable, given that `GraphECL` exclusively employs the learned `MLP` representations for inference without any reliance on input graph structures! This demonstrates that `MLP` learned by `GraphECL` is able to capture meaningful structural information that is beneficial and generalized to the downstream task.

**Inductive Setting.** To gain a better understanding of `GraphECL`'s effectiveness, we evaluate the inferred representations in a realistic production setting that encompasses both transductive and inductive nodes. In the inductive evaluation, we set aside certain test nodes (20%) from the test nodes in the transductive setting to create an inductive set (as detailed in Section C.4). We adopt the GraphSAGE (Hamilton et al., 2017a) as the GNN encoder for all GCL methods. As shown in Table 2, `GraphECL` still achieves superior or competitive performance compared to elaborate methods employing `GNN` as the inference encoder. These results support the deployment of the `MLP` learned by `GraphECL` as a significantly faster model, with minimal or no performance degradation.

## 5.2 SCALABILITY AND INFERENCE TIME COMPARISON

To further demonstrate the scalability of `GraphECL`, we evaluate the inference time on large-scale graphs (`Flickr` and `Snap-patents`). We compare the inference time with BGRL (GNN-L2W256), CCA-SSG (GNN-L2W128), and SUGRL (GNN-L1W128), where GNN-LiWj indicates the method achieving the best performance with $i$ layers of GraphSAGE with $j$ dimensions, as shown in Table 12. The MLP learned by `GraphECL` has 128 hidden dimensions. Our results in Table 12 and Figure 4 indicate that `GraphECL` achieves the highest accuracy while maintaining significantly faster

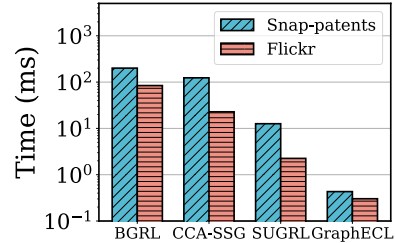

Figure 4: Inference time of different methods. Note that time axes are log-scaled.

inference times. On the large-scale graph: Snap-patents, the `MLP` learned by `GraphECL` is 286.82x faster than CCA-SSG with the same number of layers, showing the superior efficiency of `GraphECL`.

## 5.3 ABLATION STUDIES

**Ablation Studies.** We study the effects of intra-model and inter-model negative losses. We consider three ablations: **(A1)** Removing the inter-model negative pairs; **(A2)** Removing the intra-model negative pairs; and **(A3)** Removing both intra-model and inter-model negative pairs. We also explore the effects of the cross-model contrastive architecture in `GraphECL` by removing the asymmetric GNN-MLP architecture and consider other two ablations for `GraphECL`: **(A4)** using only

Table 3: Ablation studies on Flicker dataset.

| Ablation | Accuracy (%) |
|---|---|
| **A1** w/o inter-model negative pair | 42.34±0.03 |
| **A2** w/o intra-model negative pair | 42.34±0.01 |
| **A3** w/o both types of negative pairs | 40.25±0.05 |
| **A4** w/ only MLP encoder | 44.83±0.06 |
| **A1 & A4** | 42.34±0.04 |
| **A2 & A4** | 42.32±0.10 |
| **A3 & A4** | 41.28±0.02 |
| `GraphECL` | **48.49±0.15** |

the `MLP` as the encoder. Table 3 lists the results. We also find that `GraphECL` without any negative pairs performs bad, demonstrating that adding the uniformity loss of negative pairs is crucial for good generalization. Additionally, we observe that `GraphECL` with using only MLP as the encoder can not achieve the best performance, although it can also conduct fast inference, confirming the effectiveness of the designed GNN-MLP contrastive architecture. This confirms our motivation that cross-model architecture is important for capturing structural information. `GraphECL` achieves the best performance over other ablations, demonstrating that our designed components are complementary to each other.

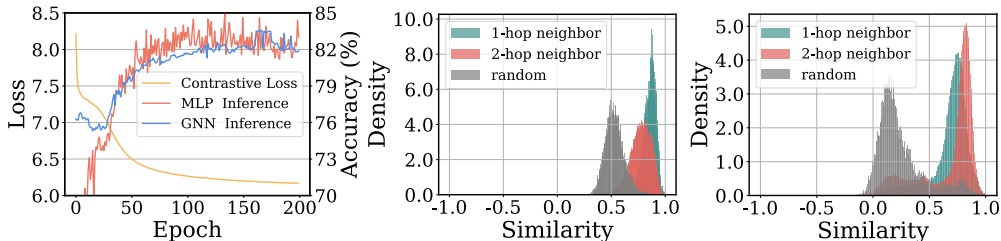

Figure 5: (Left) The training dynamics of `GraphECL` on Cora. (Right two) The pairwise cosine similarity of representations for randomly sampled node pairs, one-hop neighbors, and two-hop neighbors on Cora and Actor.

Table 4: Ablation study on the effectiveness of using generalized contrastive loss in `GraphECL`.

|  | Cora | Citeseer | Pubmed | Photo | Actor | Crocodile |
|---|---|---|---|---|---|---|
| GraphECL (InfoNCE) | 74.55±0.45 | 67.15±0.25 | 76.50±1.20 | 91.44±0.61 | 33.39±0.37 | 63.88±0.34 |
| GraphECL (Generalized) | **84.25±0.05** | **73.15±0.41** | **82.21±0.05** | **94.22±0.11** | **35.80±0.89** | **65.84±0.71** |

**Effectiveness of Generalized Contrastive Loss**. We maintain the cross-model contrastive architecture while replacing our generalized contrastive loss with the vanilla InfoNCE loss, as shown in Equation (2). Table 4 summarizes the results across all six datasets, demonstrating consistent improvements when using the generalized contrastive loss in the `GraphECL` formulation.

**Size of Negative Pairs**. We investigate the influence of different number of negative pairs (i.e., $M$) in `GraphECL` (Appendix D.2). While a proper range can lead to certain gains (Figure 7), a small number of negative samples (e.g., $M = 5$) is enough to achieve good performance.

**Representation Dimension.** We study the effects of dimensions (i.e., $K$) in Appendix D.3. Interestingly, we find that larger dimensions often yield better results, except in the case of extremely large dimensions, for both homophilic and heterophilic graphs. This observation aligns with Theorem 4.2, showing that a larger dimension can effectively reduce the upper bound of downstream errors.

**The Trade-off Parameter.** We explore the effects of the trade-off parameter of $\lambda$ which controls the balance between the two types of negative pairs. As shown in Figure 6, while a specific value can lead to certain gains, `GraphECL` is robust to different choices of the value $\lambda$ on different graphs.

### 5.4 FURTHER MODEL ANALYSIS

**Training Dynamics**. We also investigate the training process of `GraphECL`. Figure 5 (left) shows the curves of the training losses and downstream performances utilizing `GNN` and `MLP`, respectively. We can find: (1) `GraphECL` exhibits training stability, consistently enhancing performance as training losses decrease; (2) As the training proceeds, `MLP` gradually and dynamically acquires knowledge from `GNN`, facilitating the dynamic exchange of information between `GNN` and `MLP` in `GraphECL`.

**Similarity Distribution**. In addition to quantitative analysis, we visualize pairwise cosine similarities among randomly sampled nodes, one-hop neighbors, and two-hop neighbor pairs based on learned representations. Figure 5 shows that, in the homophilic graph (i.e., `Cora`), nodes exhibit similar representations to their neighbors. `GraphECL` enhances similarities between neighbor nodes compared to randomly sampled node pairs, demonstrating its ability to effectively preserve one-hop neighborhood contexts. In contrast, in the heterophilic graph (i.e., `Actor`), `GraphECL` strives to bring two-hop neighbor nodes closer together. This observation aligns seamlessly with our analytical insights, showing `GraphECL`'s aptitude for automatically capturing graph structures beyond homophily.

## 6 CONCLUSION

We present `GraphECL`, a simple and inference-efficient GCL framework for learning effective node representations from graph data. `GraphECL` introduces a cross-model contrastive architecture and a generalized contrastive loss to train a `MLP` encoder. Interestingly, `GraphECL` is significantly faster than GCL methods that require a `GNN` as the encoder, while also achieving superior performance! We demonstrate theoretically that `GraphECL` leverages neighborhood distribution as an inductive bias. Extensive experiments on various real-world graphs highlight its intriguing properties, including better inference efficiency, robustness, and generalization on both homophilic and heterophilic graphs.

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

## A  ADDITIONAL RELATED WORK

In this section, we detail comparisons with prior contrastive learning schemes on graphs. Although we have compared previous works in the main body of the paper, we further provide a clear comparison of characteristics with previous contrastive learning schemes on graphs in Table 5.

*Comparisons with GCL methods with augmentation* (Trivedi et al., 2022; Veličković et al., 2018; Zhu et al., 2021; You et al., 2020; 2022; Suresh et al., 2021; Xia et al., 2022b): This contrastive learning scheme relies on graph augmentations and is built based on the invariant assumption that the augmentation can preserve the semantic nature of samples, i.e., the augmented samples have invariant semantic labels with the original ones. This scheme requires a GNN as the encoder to achieve good performance, which is computation-intensive during inference. However, our `GraphECL` is not based on graph augmentations but directly captures the 1-hop neighborhood distribution.

*Comparisons with Graph-MLP* (Yang et al., 2016; Hu et al., 2021; Yang et al., 2021b; Liu et al., 2020): Despite its inference efficiency due to the exclusive use of `MLP`, this approach exhibits significantly lower performance compared to GCL with augmentations, as depicted in Figure 1. Moreover, this scheme over-emphasizes homophily (You et al., 2020; Xiao et al., 2022), making it difficult to generalize to graphs with heterophily (Lim et al., 2021). In contrast, our `GraphECL` enjoys good downstream performance with fast inference speed for both homophilic and heterophilic graphs.

*Comparisons with MLP-augmented GCL*: This method mentioned in Section 4.1 also relies on the invariant assumption that the augmentation can preserve the semantic nature of samples, i.e., the augmented samples have invariant semantic labels with the original ones. In addition, it suffers from significant performance degradation on downstream tasks as shown in the results in Section 4.1.

Table 5: Comparison of characteristics to previous contrastive schemes on graphs. Our `GraphECL` does not rely on invariant assumption that the augmentation can preserve the semantic nature of samples and homophily assumption that connected nodes should have similar representations.

| Contrastive Schemes | Invariant Assumption | Homophily Assumption | Task Effective | Inference Efficient |
|---|:---:|:---:|:---:|:---:|
| GCL with Augmentations | ✓ | ✗ | ✓ | ✗ |
| Graph-MLP | ✗ | ✓ | ✗ | ✓ |
| MLP-Augmented GCL | ✓ | ✗ | ✗ | ✓ |
| Our `GraphECL` | ✗ | ✗ | ✓ | ✓ |

## B  PROOFS

### B.1  PROOFS OF THEOREM 4.1

**Theorem 4.1.** *The learning dynamics w.r.t the `MLP` encoder $f_M$ with the generalized contrastive loss ($\lambda = 1$) in Equation (5) saturates when the true normalized adjacency and the estimated normalized affinity matrices agree: $\mathbf{D}^{-1}\mathbf{A} = \bar{\mathbf{D}}^{-1}\bar{\mathbf{A}}$, which implies that, for $\forall v, u \in \mathcal{V}$, we have:*

$$\mathcal{P}_n(u \mid v) = \mathcal{P}_f(u \mid v) \triangleq \frac{\exp(f_M(v)^\top f_G(u)/\tau)}{\sum_{v' \in \mathcal{V}} \exp(f_M(v)^\top f_G(v')/\tau)}, \tag{8}$$

*where $\mathcal{P}_n(u \mid v)$ is the 1-hop neighborhood distribution (i.e., the v-th row of the normalized adjacency matrix) and $\mathcal{P}_f(u \mid v)$ is the Boltzmann distribution estimated by encoders with temperature $\tau$.*

*Proof.* We first show that minimizing `GraphECL` objective with $\lambda = 1$ is approximately to minimizing the losses of the positive and negative pairs on `MLP` representations.

$$\begin{aligned}
\mathcal{L}_{\text{ECL}} &= -\frac{1}{|\mathcal{V}|} \sum_{v \in \mathcal{V}} \frac{1}{|\mathcal{N}(v)|} \sum_{u \in \mathcal{N}(v)} \log \frac{\exp(f_M(v)^\top f_G(u)/\tau)}{\sum_{v^- \in \mathcal{V}} \exp(f_G(u)^\top f_G(v^-)/\tau) + \exp(f_M(v)^\top f_G(v^-)/\tau)}, \\
&\geq \frac{1}{|\mathcal{V}|} \sum_{v \in \mathcal{V}} \left( \frac{1}{|\mathcal{N}(v)|} \sum_{u \in \mathcal{N}(v)} -f_M(v)^\top f_G(u)/\tau + \log \sum_{v^- \in \mathcal{V}} \exp(f_M(v)^\top f_G(v^-)/\tau) \right) \\
&= \underbrace{\frac{1}{|\mathcal{V}|} \sum_{v \in \mathcal{V}} \frac{1}{|\mathcal{N}(v)|} \sum_{u \in \mathcal{N}(v)} -f_M(v)^\top f_G(u)/\tau}_{\mathcal{L}_{pos}} + \underbrace{\frac{1}{|\mathcal{V}|} \sum_{v \in \mathcal{V}} \log \sum_{v^- \in \mathcal{V}} \exp(f_M(v)^\top f_G(v^-)/\tau)}_{\mathcal{L}_{neg}}. \tag{9}
\end{aligned}$$

Then, we consider the unfolded iterations of descent steps on MLP representation $f_M(v)$. Specifically, we first consider taking the derivatives of $\mathcal{L}_{pos}$ and $\mathcal{L}_{neg}$ on $f_M(v)$:

$$\frac{\partial \mathcal{L}_{pos}}{\partial f_M(v)} = -\frac{1}{|\mathcal{V}|}\frac{1}{|\mathcal{N}(v)|}\sum_{u\in\mathcal{N}(v)} f_G(u)/\tau, \quad \frac{\partial \mathcal{L}_{neg}}{\partial f_M(v)} = \frac{1}{|\mathcal{V}|}\sum_{v^-\in\mathcal{V}} \frac{\exp(f_M(v)^\top f_G(v^-)/\tau)f_G(v^-)/\tau}{\sum_{\tilde{v}^-\in\mathcal{V}}\exp(f_M(v)^\top f_G(\tilde{v}^-)/\tau)}. \quad (10)$$

As we denote representation matrix as $\mathbf{M}$ with $f_M(v)$ as the $v$-th row, the Equation (10) can be written as the following matrix forms for simplicity and clarity:

$$\frac{\partial \mathcal{L}_{pos}}{\partial \mathbf{M}} = -\frac{1}{|\mathcal{V}|}\frac{1}{\tau}\mathbf{D}^-\mathbf{A}\mathbf{G}, \quad \frac{\partial \mathcal{L}_{neg}}{\partial \mathbf{M}} = \frac{1}{|\mathcal{V}|}\frac{1}{\tau}\bar{\mathbf{D}}^{-1}\bar{\mathbf{A}}\mathbf{G}, \quad (11)$$

where $\bar{\mathbf{A}} = \exp(\mathbf{M}\mathbf{G}^\top/\tau)$ is the affinity matrix based on feature similarity. $\bar{\mathbf{D}} = \deg(\bar{\mathbf{A}})$ is the diagonal matrix, whose element in the $v$-th row and $v$-th column is the sum of the $v$-th row of $\bar{\mathbf{A}}$. In order to reduce the losses on positive pairs and negative pairs, we can take a single step performing gradient descent, which is to update MLP representations $\mathbf{M}$ as follows:

$$\mathbf{M}^{(t+1)} = \mathbf{M}^{(t)} - \alpha\frac{\partial(\mathcal{L}_{pos} + \mathcal{L}_{neg})}{\partial \mathbf{M}} = \mathbf{M}^{(t)} - \frac{\alpha}{\tau}(\bar{\mathbf{D}}^{-1}\bar{\mathbf{A}} - \mathbf{D}^-\mathbf{A})\mathbf{G}, \quad (12)$$

where $\mathbf{M}^{(t+1)}$ and $\mathbf{M}^{(t)}$ denote the representations before and after the update, respectively, and $\alpha > 0$ is the step size of the gradient descent. Note that the constant $1/|\mathcal{V}|$ has been absorbed in $\alpha$. We can easily notice the updating in Equation (12) reveals the stationary point, i.e, global minimum, of the learning: $\bar{\mathbf{D}}^{-1}\bar{\mathbf{A}} = \mathbf{D}^-\mathbf{A}$. Combining this with Equation (9) completes the proof. $\qquad\square$

### B.2 PROOFS OF THEOREM 4.2

To prove Theorem 4.2, we first present the following Lemma:

**Lemma B.1.** *(Theorem B.3 (page 32) in (HaoChen et al., 2021)). Let $f^*$ be a minimizer of the spectral contrastive loss: $\mathcal{L}_{SCL} = \sum_{x,x'\in\mathcal{X}} -2 \cdot w_{xx'} \cdot f(x)^\top f(x') + w_x w_{x'} \cdot (f(x)^\top f(x'))^2$, where $w_{x,x'} = w(x)w(x'|x)$ is the probability of a random positive pair being $(x, x')$ while $w_x$ is the probability of a randomly selected data point being $x$, we have:*

$$\mathcal{E}(f^*) \triangleq \min_W \sum_{x\in\mathcal{X}} w_x \cdot \mathbb{1}[g_{f^*,W}(x) \neq y(x)] \leq \frac{\phi^{\hat{y}}}{\lambda_{k+1}}, \quad (13)$$

*where $\phi^{\hat{y}} = \sum_{x,x'\in\mathcal{X}} w_{xx'} \cdot \mathbb{1}[\hat{y}(x) \neq \hat{y}(x')]$ and $W$ is the downstream linear classifier. $\lambda_{k+1}$ is the $(K+1)$-th smallest eigenvalues of the matrix: $\mathbf{I} - \mathbf{D}^{-1/2}\mathbf{P}\mathbf{D}^{-1/2}$ where $\mathbf{P} \in \mathbb{R}^{N\times N}$ is a symmetric probability matrix with $\mathbf{P}_{xx'} = w_{xx'}$ and $\mathbf{D}_{\mathbf{P}} \in \mathbb{R}^{N\times N}$ is a diagonal matrix with $(\mathbf{D}_{\mathbf{P}})_{xx} = w_x$.*

We also introduce the following Lemma which asserts that multiplying the embedding matrix on the right by an invertible matrix does not affect the linear probing error.

**Lemma B.2.** *(Lemma 3.1 (page 8) in (HaoChen et al., 2021)). Consider an embedding matrix $\mathbf{F} \in \mathbb{R}^{N\times k}$ and a linear classifier $\mathbf{B} \in \mathbb{R}^{k\times r}$. Let $\mathbf{D} \in \mathbb{R}^{N\times N}$ be a diagonal matrix with positive diagonal entries and $\mathbf{Q} \in \mathbb{R}^{k\times k}$ be an invertible matrix. Then, for any matrix $\widetilde{\mathbf{F}} = \mathbf{D} \cdot \mathbf{F} \cdot \mathbf{Q}$, the linear classifier $\widetilde{\mathbf{B}} = \mathbf{Q}^{-1}\mathbf{B}$ on $\widetilde{\mathbf{F}}$ has the same prediction as $\mathbf{B}$ on $\mathbf{F}$. Thus, we have $\mathcal{E}(\mathbf{F}) = \mathcal{E}(\widetilde{\mathbf{F}})$.*

Intuitively, this Lemma suggests that although there might not be a single unique optimal solution, when we employ the representation within the context of linear probing, the linear classifier can efficiently handle variations caused by affine transformations. Consequently, it produces identical classification errors across different variants when operating at the optimal settings.

**Theorem 4.2.** *Let $f_M^*$ be the global minimum of generalized contrastive loss ($\lambda = 1$) in Equation (5) and $y(v)$ denote the label of $v$. $\sigma_1 \geq \cdots \geq \sigma_N$ are the eigenvalues with descending order of the normalized adjacency matrix $\mathbf{D}^{-1}\mathbf{A}$. Then, the linear probing error of $f_M^*$ is upper-bounded by:*

$$\mathcal{E}(f_M^*) \triangleq \min_W \frac{1}{|\mathcal{V}|}\sum_{v\in\mathcal{V}} \mathbb{1}[g_{f^*,W}(v) \neq y(v)] \leq \frac{1-\alpha}{1-\sigma_{K+1}}, \quad (14)$$

*where $\alpha = \frac{1}{|\mathcal{V}|}\sum_{v\in\mathcal{V}}\frac{1}{|\mathcal{N}(v)|}\sum_{u\in\mathcal{N}(v)}\mathbb{1}[y(v) = y(u)]$ and $K$ is the dimension of the representation.*

*Proof.* Based on Equation (9), we further have the following:

$$\mathcal{L}_{\text{ECL}} \geq \frac{1}{|\mathcal{V}|} \sum_{v \in \mathcal{V}} (\frac{1}{|\mathcal{N}(v)|} \sum_{u \in \mathcal{N}(v)} -f_M(v)^\top f_G(u)/\tau + \log \sum_{v^- \in \mathcal{V}} \exp(f_M(v)^\top f_G(v^-)/\tau)).$$

$$= \frac{1}{|\mathcal{V}|} \sum_{v \in \mathcal{V}} (\frac{1}{|\mathcal{N}(v)|} \sum_{u \in \mathcal{N}(v)} -f_M(v)^\top f_G(u)/\tau + \log \sum_{v^- \in \mathcal{V}} \frac{\exp(f_M(v)^\top f_G(v^-)/\tau)}{|\mathcal{V}|} |\mathcal{V}|)$$

$$= \frac{1}{|\mathcal{V}|} \sum_{v \in \mathcal{V}} (\frac{1}{|\mathcal{N}(v)|} \sum_{u \in \mathcal{N}(v)} -f_M(v)^\top f_G(u)/\tau + \log \sum_{v^- \in \mathcal{V}} \frac{\exp(f_M(v)^\top f_G(v^-)/\tau)}{|\mathcal{V}|} + \log |\mathcal{V}|)$$

$$\overset{c}{=} \frac{1}{|\mathcal{V}|} \sum_{v \in \mathcal{V}} (\frac{1}{|\mathcal{N}(v)|} \sum_{u \in \mathcal{N}(v)} -f_M(v)^\top f_G(u)/\tau + \log \sum_{v^- \in \mathcal{V}} \frac{\exp(f_M(v)^\top f_G(v^-)/\tau)}{|\mathcal{V}|})$$

$$\geq \frac{1}{|\mathcal{V}|} \sum_{v \in \mathcal{V}} (\frac{1}{|\mathcal{N}(v)|} \sum_{u \in \mathcal{N}(v)} -f_M(v)^\top f_G(u)/\tau + \sum_{v^- \in \mathcal{V}} \frac{1}{|\mathcal{V}|} (f_M(v)^\top f_G(v^-)/\tau)) \tag{15}$$

$$= \frac{1}{|\mathcal{V}|} \sum_{v \in \mathcal{V}} \frac{1}{|\mathcal{N}(v)|} \sum_{u \in \mathcal{N}(v)} -f_M(v)^\top f_G(u)/\tau + \frac{1}{|\mathcal{V}|} \frac{1}{|\mathcal{V}|} \sum_{v \in \mathcal{V}} \sum_{v^- \in \mathcal{V}} f_M(v)^\top f_G(v^-)/\tau$$

$$\geq \frac{1}{|\mathcal{V}|} \sum_{v \in \mathcal{V}} \frac{1}{|\mathcal{N}(v)|} \sum_{u \in \mathcal{N}(v)} -f_M(v)^\top f_G(u)/\tau + \frac{1}{|\mathcal{V}|} \frac{1}{|\mathcal{V}|} \sum_{v \in \mathcal{V}} \sum_{v^- \in \mathcal{V}} (f_M(v)^\top f_G(v^-))^2/\tau \tag{16}$$

$$\overset{c}{=} \frac{1}{|\mathcal{V}|} \sum_{v \in \mathcal{V}} \frac{1}{|\mathcal{N}(v)|} \sum_{u \in \mathcal{N}(v)} -2f_M(v)^\top f_G(u) + \frac{1}{|\mathcal{V}|} \frac{1}{|\mathcal{V}|} \sum_{v \in \mathcal{V}} \sum_{v^- \in \mathcal{V}} (f_M(v)^\top f_G(v^-))^2 \triangleq \mathcal{L}_{cross}, \tag{17}$$

where the symbol $\overset{c}{=}$ indicates equality up to a multiplicative and/or additive constant. Here, we utilize Jensen's inequality in (15). Inequality (16) holds because $f_M(v)$ and $f_G(u)$ are $\ell_2$-normalized, and we consider the embedding heads consisting of last-layer ReLU neural networks. We define the two metrics $\tilde{\mathbf{M}}$ and $\tilde{\mathbf{G}}$. Equation (17) holds if we set the temperature of positive pairs is twice to it of negative pairs. Here $(\tilde{\mathbf{M}})_v = |\mathcal{V}|^{-1/2} f_M(v)$ and $(\tilde{\mathbf{G}})_u = |\mathcal{V}|^{-1/2} f_G(u)$. Then. the loss in Equation (17) is equivalent to the low-rank asymmetric matrix factorization loss up to a constant:

$$\mathcal{L}_{\text{AMF}} = \|\mathbf{D}^{-1}\mathbf{A} - \tilde{\mathbf{M}}\tilde{\mathbf{G}}^\top\| = \mathcal{L}_{cross} + const \tag{18}$$

According to Eckart–Young–Mirsky theorem (Eckart & Young, 1936), the optimal solution $\tilde{\mathbf{M}}^*$ and $\tilde{\mathbf{G}}^*$ of $\mathcal{L}_{\text{AMF}}$ can be respectively represented as follows:

$$\tilde{\mathbf{M}}^*(\tilde{\mathbf{G}}^*)^\top = \mathbf{U}^K \deg(\sigma_1, \dots, \sigma_K)(\mathbf{V}^K)^\top \tag{19}$$

where we denote $\mathbf{D}^{-1}\mathbf{A}\mathbf{U}\Sigma\mathbf{V}^\top$ as the spectral decomposition of $\mathbf{D}^{-1}\mathbf{A}$. $(\sigma_1, \dots, \sigma_K)$ are the $K$-largest eigenvalue of $\mathbf{D}^{-1}\mathbf{A}$. The $k$-th column of $\mathbf{U}^K \in \mathbb{R}^{|\mathcal{V}| \times K}$ is the corresponding eigenvector of the $k$-th largest eigenvalue and $\mathbf{V}^K \in \mathbb{R}^{|\mathcal{V}| \times K}$ is a unitary matrix. Then the optimal solution $\tilde{\mathbf{M}}^*$ and $\tilde{\mathbf{G}}^*$ can be represented as follows:

$$\tilde{\mathbf{M}}^* = \mathbf{U}^K \mathbf{B} \mathbf{R}, \quad \tilde{\mathbf{G}}^* = \mathbf{V}^K \deg(\sigma_1, \dots, \sigma_K)\mathbf{B}^{-1}\mathbf{R}, \tag{20}$$

where $\mathbf{R} \in \mathbb{R}^{K \times K}$ is a unitary matrix and $\mathbf{B}$ is an invertible diagonal matrix. Since $(\tilde{\mathbf{M}})_v = |\mathcal{V}|^{-1/2} f_M(v)$ and $(\tilde{\mathbf{G}})_u = |\mathcal{V}|^{-1/2} f_G(u)$, we have:

$$f_M^*(v) = |\mathcal{V}|^{1/2}((\mathbf{U}^K)_v \mathbf{B} \mathbf{R})^\top, \quad f_G^*(u) = |\mathcal{V}|^{1/2}((\mathbf{V}^K)_u \deg(\sigma_1, \dots, \sigma_K)\mathbf{B}^{-1}\mathbf{R})^\top. \tag{21}$$

Similar, if we consider optimizing following uni-model spectral contrastive loss:

$$\mathcal{L}_{uni} = \frac{1}{|\mathcal{V}|} \sum_{v \in \mathcal{V}} \frac{1}{|\mathcal{N}(v)|} \sum_{u \in \mathcal{N}(v)} -2f_M(v)^\top f_M(u) + \frac{1}{|\mathcal{V}|} \frac{1}{|\mathcal{V}|} \sum_{v \in \mathcal{V}} \sum_{v^- \in \mathcal{V}} (f_M(v)^\top f_M(v^-))^2. \tag{22}$$

The optimal solution $\hat{f}_M^*$ of this uni-model spectral contrastive loss can be represented as follow:

$$\hat{f}_M^*(v) = |\mathcal{V}|^{1/2}((\mathbf{U}_{uni}^K)_v \mathbf{B}_{uni} \mathbf{R}_{uni})^\top. \tag{23}$$

Since one can easily find the uni-model spectral contrastive loss in Equation (22) also decomposes the matrix $\mathbf{D}^{-1}\mathbf{A}$, the $\mathbf{U}_{uni}^K = \mathbf{U}^K$. As $\mathbf{B}, \mathbf{R}, \mathbf{B}_{uni}, \mathbf{R}_{uni}$ are invertible matrices and the product of the invertible matrices is still invertible, we have:

$$f_M^*(v) = \hat{f}_M^*(v)\mathbf{T}, \tag{24}$$

where $\mathbf{T} = (\mathbf{B}_{uni})^{-1}(\mathbf{R}_{uni})^{-1}\mathbf{BR}$. With Lemma B.2, we establish that $\mathcal{E}(f_M^*) = \mathcal{E}(\hat{f}_M^*)$. Additionally, we observe that the loss in Equation (22) shares the same form as the spectral contrastive loss when we define $\frac{1}{|\mathcal{V}|}\mathbf{D}^{-1}\mathbf{A} = \hat{\mathbf{A}}$ i.e., $w_x = \frac{1}{|\mathcal{V}|}$ and $w_{x'|x} = (\mathbf{D}^{-1}\mathbf{A})_{x,x'}$. It's worth noting that $\mathbf{D}^{-1}\mathbf{A} = \mathbf{D}^{-1/2}\mathbf{A}\mathbf{D}^{-1/2}$ forms a symmetric matrix due to our random sampling process, which ensures that the same neighbors are sampled for each central node, approximately resulting in equal node degrees. Thus, with Lemma B.1, we can obtain the following:

$$\mathcal{E}(f_M^*) = \mathcal{E}(\hat{f}_M^*) \triangleq \min_W \frac{1}{|\mathcal{V}|}\sum_{v \in \mathcal{V}} \mathbb{1}[g_{f^*,W}(v) \neq y(v)] \leq \frac{1 - \alpha}{1 - \sigma_{K+1}} \qquad (25)$$

where $\alpha = 1/|\mathcal{V}|\sum_{v \in \mathcal{V}} \frac{1}{|\mathcal{N}(v)|}\sum_{u \in \mathcal{N}(v)} \mathbb{1}[y(v) = y(u)]$ and $\sigma_{K+1}$ is the $(K+1)$-th largest singular value of the normalized adjacency matrix $\mathbf{D}^{-1}\mathbf{A}$. Given the above, the proof is finished. □

## C   EXPERIMENTAL DETAILS

Table 6: Statistics of Benchmark Datasets.

| | Cora | Citeseer | Pubmed | Photo | Flickr | Crocodile | Actor | Wisconsin | Cornell | Texas | Snap-patents |
|---|---|---|---|---|---|---|---|---|---|---|---|
| #Nodes | 2,708 | 3,327 | 19,717 | 7,650 | 89,250 | 11,631 | 7,600 | 251 | 183 | 183 | 2,923,922 |
| #Edges | 5,278 | 4,552 | 44,324 | 119,081 | 899,756 | 360,040 | 33,544 | 466 | 295 | 309 | 13,975,788 |
| #Classes | 7 | 6 | 3 | 8 | 7 | 5 | 5 | 5 | 5 | 5 | 5 |
| #Features | 1,433 | 3,703 | 500 | 745 | 500 | 2,089 | 931 | 1,703 | 1,703 | 1,793 | 269 |
| $\mathcal{H}(\mathcal{G})$ | 0.83 | 0.71 | 0.79 | 0.85 | 0.32 | 0.30 | 0.22 | 0.16 | 0.11 | 0.06 | 0.22 |
| $\mathcal{S}(\mathcal{G})$ | 0.89 | 0.81 | 0.87 | 0.91 | 0.33 | 0.71 | 0.68 | 0.42 | 0.40 | 0.79 | 0.29 |

### C.1   ONE-HOP NODE HOMOPHILY LEVEL

We use the node homophily ratio to measure the one-hop neighbor homophily of the graph (Pei et al., 2020). The detailed node homophily ratio $\mathcal{H}(\mathcal{G})$ can be computed as:

$$\mathcal{H}(\mathcal{G}) = \frac{1}{|\mathcal{V}|}\sum_{v \in \mathcal{V}}\frac{1}{\mathcal{N}(v)}\sum_{u \in \mathcal{N}(v)} \mathbb{1}(y(v) = y(u)). \qquad (26)$$

### C.2   NEIGHBORHOOD CONTEXT SIMILARITY

To validate the assumption that nodes belonging to an identical semantic category are likely to exhibit similar patterns in their one-hop neighborhoods, even in heterophilic graphs, we examine whether nodes with the same label demonstrate similar distributions of labels in their neighborhoods regardless of the homophily. We evaluate the characteristic by computing the class neighborhood similarity (Ma et al., 2022), which is defined as:

$$s\left(m, m'\right) = \frac{1}{|\mathcal{V}_m||\mathcal{V}_{m'}|}\sum_{u \in \mathcal{V}_m, v \in \mathcal{V}_{m'}} \cos(d(u), d(v)), \qquad (27)$$

where $M$ denotes the total number of classes, $\mathcal{V}_m$ represents the set of nodes classified as $m$, and $d(u)$ is the empirical histogram of the labels of node $u$'s neighbors across $M$ classes. The cosine similarity function is represented by $\cos(\cdot)$. This metric for cross-class neighborhood similarity quantifies the differences in neighborhood distributions between varying classes. When $m = m', s\left(m, m'\right)$ determines the intra-class similarity. To quantify the neighborhood similarity, we take the average of the intra-class similarities across all classes:

$$\mathcal{S}(\mathcal{G}) = \sum_{m=1}^{M}\frac{1}{M}s(m, m). \qquad (28)$$

If nodes with identical labels exhibit similar neighborhood distributions, then the class neighborhood similarity $\mathcal{S}(\mathcal{G})$ will be high. As shown in Table 6, we can observe that many heterophilic graphs exhibit stronger neighborhood similarity, even when the homophily ratio is low.

## C.3 DATASETS DETAILS

The statistics of benchmark datasets, including homophily levels and 1-hop neighborhood similarities, are given in Table 6. All datasets and public splits can found in PyTorch Geometric: `https://pytorch-geometric.readthedocs.io/en/latest/modules/datasets.html`.

**Cora, Citeseer, and Pubmed.** (Kipf & Welling, 2016a; Yang et al., 2016) These datasets serve as some of the most prevalent benchmarks for node classification. Each one constitutes a high-homophily graph representing citations, with nodes symbolizing documents and edges depicting citation relationships between them. The classification of each node is determined by the respective research field. Features of the nodes are derived from a bag of words model applied to their abstracts. We utilize the public split: a fixed 20 nodes from each class for training and another distinct set of 500 and 1000 nodes for validation and testing, respectively (Kipf & Welling, 2016a).

**Photo.** (Thakoor et al., 2022; McAuley et al., 2015) The graphs originate from the Amazon co-purchase graph (McAuley et al., 2015), where nodes denote products and edges connect pairs of items often bought together. In the context of Photo dataset, products are categorized into 8 classes based on their category, and the node features are represented by a bag-of-words model of the product's reviews. We employ a random splits of the nodes into training, validation, and testing sets, following a 10/10/80% ratio respectively following (Thakoor et al., 2022).

**Flickr.** (Zeng et al., 2020) In this graph, each node symbolizes an individual image uploaded to Flickr. An edge is established between the nodes of two images if they share certain attributes, such as geographic location, gallery, or user comments. The node features are represented by a 500-dimensional bag-of-word model provided by NUS-wide. Regarding labels, we examined the 81 tags assigned to each image and manually consolidated them into 7 distinct classes, with each image falling into one of these categories. We utilize a random node division method, adhering to a 50/25/25% split for training, validation, and testing sets following (Zeng et al., 2020).

**Cornell, Wisconsin and Texas.** (Pei et al., 2020) These are networks of webpages, gathered from the computer science departments of various universities by Carnegie Mellon University. In each network, the nodes represent individual webpages, while the edges signify hyperlinks between them. The features of the nodes are depicted using bag-of-words representations of the webpages. The objective is to categorize each node into one of five classes: student, project, course, staff, or faculty.

**Actor.** (Pei et al., 2020) This is a subgraph induced solely by actors, derived from the broader film-director-actor-writer network. In this subgraph, nodes represent actors, while edges denote the co-occurrence of two nodes on the same Wikipedia page. The features of the nodes are constituted by keywords found on Wikipedia pages. Labels are categorized into five groups based on the content of the actor's corresponding Wikipedia page.

**Crocodile.** (Rozemberczki et al., 2021) These networks are sourced from Wikipedia, with nodes symbolizing web pages and edges denoting the hyperlinks between them. The features of the nodes consist of various informative nouns found on the Wikipedia pages. The labels assigned to the nodes are determined by the average daily traffic each web page receives.

For **Texas, Wisconsin, Cornell, Crocodile, Actor and Snap-patents**, we use the raw data provided by the Geom-GCN (Pei et al., 2019) with the standard fixed 10-fold split for our experiment.

**Snap-patents.** (Lim et al., 2021) The snap-patents dataset encompasses a collection of utility patents from the US, where each node represents a patent, and edges are formed between patents that cite one another. The features of the nodes are extracted from the metadata of the patents. In this work, we introduce a task aiming to predict the time at which a patent was granted, which is categorized into five classes. We utilize the unprocessed data from Lim et al. (2021), employing the standard 10-fold split for our experimental setup.

## C.4 TRANSDUCTIVE AND INDUCTIVE SETTINGS FOR UNSUPERVISED REPRESENTATION LEARNING

**Transductive Setting**. To fully evaluate the model, we consider two settings: transductive (tran) and inductive (ind). In the transductive setting, our evaluation consists of two phases. Initially, we pre-train models on graph $\mathcal{G}$, followed by the generation of representations for all nodes within the

Table 7: Node clustering performance in terms of NMI (%) on homophilic and heterophilic graphs

| Method | Citeseer | Cora | Photo | Texas | Cornell | Actor |
|--------|----------|------|-------|-------|---------|-------|
| VGAE | $36.40_{\pm 0.01}$ | $38.92_{\pm 0.02}$ | $53.00_{\pm 0.04}$ | $27.75_{\pm 0.16}$ | $17.87_{\pm 0.13}$ | $15.92_{\pm 0.07}$ |
| DGI | $43.90_{\pm 0.00}$ | $31.80_{\pm 0.02}$ | $47.60_{\pm 0.03}$ | $34.17_{\pm 0.07}$ | $15.92_{\pm 0.15}$ | $19.31_{\pm 0.05}$ |
| BGRL | $45.38_{\pm 0.04}$ | $\underline{47.35}_{\pm 0.03}$ | $54.61_{\pm 0.08}$ | $33.59_{\pm 0.15}$ | $19.74_{\pm 0.14}$ | $22.03_{\pm 0.06}$ |
| DSSL | $\underline{45.91}_{\pm 0.06}$ | $46.77_{\pm 0.04}$ | $\underline{54.99}_{\pm 0.05}$ | $38.22_{\pm 0.15}$ | $\underline{20.36}_{\pm 0.08}$ | $21.32_{\pm 0.10}$ |
| SUGRL | $45.86_{\pm 0.08}$ | $45.66_{\pm 0.04}$ | $53.05_{\pm 0.05}$ | $\underline{39.41}_{\pm 0.10}$ | $19.55_{\pm 0.05}$ | $\underline{24.69}_{\pm 0.17}$ |
| GraphECL | $\mathbf{46.87}_{\pm 0.06}$ | $\mathbf{48.52}_{\pm 0.07}$ | $\mathbf{56.71}_{\pm 0.05}$ | $\mathbf{44.56}_{\pm 0.02}$ | $\mathbf{27.85}_{\pm 0.12}$ | $\mathbf{29.71}_{\pm 0.20}$ |

graph, denoted as $z_v$ for $v \in \mathcal{V}$. Subsequently, we employ a linear classifier trained on the fixed learned representations using labeled data $\boldsymbol{Z}^L$ and $\boldsymbol{Y}^L$. Finally, we assess the remaining inferred representations $\boldsymbol{Z}^U$ with corresponding labels $\boldsymbol{Y}^U$.

**Inductive Setting**. In the unsupervised inductive setting, we randomly select 20% of the nodes as a test set for inductive evaluation. Specifically, we partition the unlabeled nodes $\mathcal{V}^U$ into two separate subsets: observed and inductive (i.e., $\mathcal{V}^U = \mathcal{V}^U_{\text{obs}} \cup \mathcal{V}^U_{\text{ind}}$). This leads to the creation of three distinct graphs: $\mathcal{G} = \mathcal{G}^L \cup \mathcal{G}^U_{\text{obs}} \cup \mathcal{G}^U_{\text{ind}}$, where no nodes are shared between $\mathcal{G}^L \cup \mathcal{G}^U_{\text{obs}}$ and $\mathcal{G}^U_{\text{ind}}$. Importantly, during self-supervised training, we remove the edges connecting $\mathcal{G}^L \cup \mathcal{G}^U_{\text{obs}}$ and $\mathcal{G}^U_{\text{ind}}$. Upon completing the self-supervised pre-training on $\mathcal{G}^L \cup \mathcal{G}^U_{\text{obs}}$, we generate node representations for all nodes. Consequently, the learned representations and associated labels are partitioned into three separate sets: $\boldsymbol{Z} = \boldsymbol{Z}^L \cup \boldsymbol{Z}^U_{obs} \cup \boldsymbol{Z}^U_{\text{ind}}$ and $\boldsymbol{Y} = \boldsymbol{Y}^L \cup \boldsymbol{Y}^U_{obs} \cup \boldsymbol{Y}^U_{\text{ind}}$. A downstream classifier is then trained on the fixed learned representations using labeled data $\boldsymbol{Z}^L$ and $\boldsymbol{Y}^L$. Finally, we evaluate the remaining representations $\boldsymbol{Z}^U$ and $\boldsymbol{Z}^U_{\text{ind}}$ on downstream classifier with labels $\boldsymbol{Y}^U_{obs}$ and $\boldsymbol{Y}^U_{\text{ind}}$, respectively.

## C.5 SETUP AND HYPER-PARAMETER SETTINGS

We utilized the official implementations publicly released by the authors for the baseline methods. To ensure a fair comparison, we conducted a grid search to determine the optimal hyperparameters for these baselines. Our experiments were conducted on a machine equipped with an NVIDIA RTX A6000 GPU boasting 48GB of GPU memory. For all experiments, we employed the Adam optimizer (Kingma & Ba, 2014). A small-scale grid search was employed to select the best hyperparameters for all methods. Specifically, for our GraphECL approach, we explored the following hyperparameter ranges: $\lambda$ from {0.001, 0.01, 0.1, 0.5, 1}, $K$ from {256, 512, 1024, 2048, 4096}, $\tau$ from {0.5, 0.75, 0.99, 1}, and the number of negative pairs $M$ from {1, 5, 10} when negative sampling was utilized. In addition, we tuned the learning rate from the set {0.001, 0.0005, 0.0001} and the weight decay from the set {0, 0.0001, 0.0003, 0.000001}. The selection of the optimal hyperparameter configuration was based solely on the average accuracy on the validation set.

## D ADDITIONAL EXPERIMENTAL RESULTS

### D.1 NODE CLUSTERING PERFORMANCE

We also perform node clustering to assess the quality of learned node representations. Specifically, we acquire node representations using GraphECL and subsequently apply $k$-means clustering to these representations, setting the number of clusters equal to the number of ground truth classes. This experiment is repeated five times, and the average normalized mutual information (NMI) for clustering is reported in Table 7 for both homophilic and heterophilic graphs.

From the table, it is evident that our GraphECL consistently enhances node clustering performance when compared to state-of-the-art self-supervised learning baselines across eight datasets. This observation, coupled with the node classification results, underscores the effectiveness of GraphECL in acquiring more expressive and resilient node representations for a variety of downstream tasks. These findings further validate that modeling one-hop neighborhood patterns confers advantages to downstream tasks on real-world graphs with varying degrees of homophily.

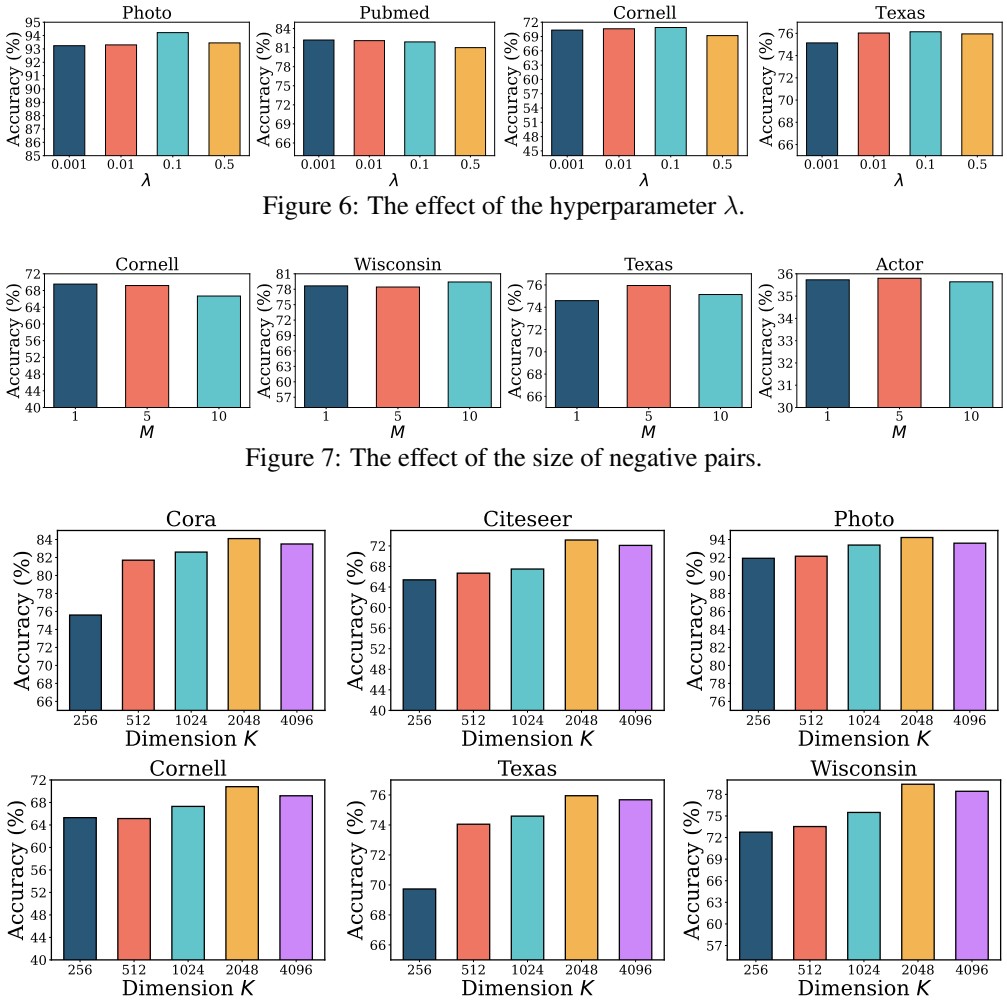

Figure 6: The effect of the hyperparameter $\lambda$.

Figure 7: The effect of the size of negative pairs.

Figure 8: The effect of the dimensions of representations.

## D.2 THE EFFECT OF SIZE OF NEGATIVE PAIRS

We conducted a sweep over the size of negative samples, denoted as $M$, to study its impact on performance. We varied $M$ across the values $1, 5, 10$. For each $M$ value, we first learned node representations and subsequently applied these learned representations to node classification. The results of this experiment are shown in Figure 7. From the figure, we observe that even a small number of negative samples, such as $M = 5$, is sufficient to achieve good performance across all graphs, demonstrating that `GraphECL` is particularly robust to reduced negative pairs.

## D.3 THE EFFECT OF REPRESENTATION DIMENSION

We investigate the impact of different dimensions of representations. Figures 8 shows the node classification results with varying dimensions on homophily and heterophilic graphs. From the figure, we can observe that larger dimensions often yield better results for both homophilic and heterophilic graphs. This observation aligns with Theorem 4.2, which shows that a larger dimension can effectively reduce the upper bound of downstream errors. Training with extremely large dimensions for some graphs may lead to a slight drop of performance as `GraphECL` may suffer from the over-fitting issue.

## D.4 MORE SIMILARITY HISTOGRAMS OF REPRESENTATIONS

Figure 9 presents additional results on representation similarity. These observations align with the findings in the main body of the paper. As shown in Figure 9, we notice that randomly sampled node

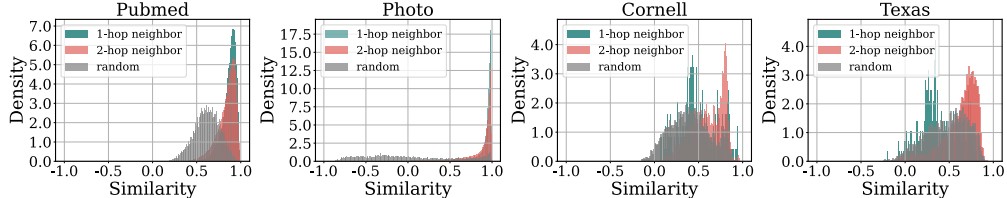

Figure 9: The distribution of pair-wise cosine similarity of learned representations on randomly sampled node pairs, one-hop neighbors and two-hop neighbors.

Table 8: Additional comparison with CGD (Zheng et al., 2022), SimGCL (Yu et al., 2022) and FastGCL (Wang et al., 2022b) under the transductive setting on benchmarking homophilic and heterophilic graphs. The best and runner up methods are marked with boldface and underline, respectively.

| Datasets | Cora | Citeseer | Pubmed | Photo | Flickr | Cornell | Wisconsin | Texas | Crocodile | Actor | Snap-patents |
|---|---|---|---|---|---|---|---|---|---|---|---|
| GGD | 83.90±0.40 | 73.00±0.60 | 81.30±0.80 | 92.50±0.60 | 46.33±0.20 | 52.98±1.32 | 61.85±0.43 | 59.36±1.23 | 57.57±0.39 | 28.27±0.23 | 24.38±0.57 |
| SimGCL | 83.01±0.32 | 72.05±0.29 | 80.57±0.52 | 91.55±0.12 | 46.85±0.19 | 53.52±0.71 | 60.55±0.32 | 58.91±0.88 | 58.51±0.19 | 28.63±0.33 | 25.24±0.21 |
| FastGCL | 82.33±0.51 | 71.60±0.51 | 80.41±0.62 | 92.91±0.07 | 45.21±0.11 | 51.35±0.92 | 57.74±0.28 | 57.72±0.76 | 55.23±0.25 | 27.71±0.15 | 24.05±0.31 |
| GraphECL | **84.25±0.05** | **73.15±0.41** | **82.21±0.05** | **94.22±0.11** | **48.49±0.15** | **69.19±6.86** | **79.41±2.19** | **75.95±5.33** | **65.84±0.71** | **35.80±0.89** | **27.22±0.06** |

Table 9: Node classification results (%) under the transductive setting on additional graphs. The best and runner up methods are marked with boldface and underline, respectively. From the table, we can observe that our simple GraphECL performs well on the additional graphs and achieves better (or competitive) performance compared to baselines with GCN, which further strengthens our contribution.

| Datasets | WikiCS | Computers | CS | Physics | Ogbn-arxiv | Ogbn-products |
|---|---|---|---|---|---|---|
| GCA | 78.35±0.05 | 88.94±0.15 | 93.10±0.01 | 95.70±0.04 | 68.20±0.20 | 78.96±0.15 |
| SUGRL | 78.83±0.31 | 88.90±0.20 | 92.83±0.23 | 95.38±0.11 | 69.30±0.20 | 82.60±0.40 |
| BGRL | 79.98±0.10 | **90.34±0.19** | 93.31±0.13 | 95.73±0.05 | 71.64±0.12 | 81.32±0.21 |
| CCA-SSG | 79.31±0.21 | 88.74±0.28 | 93.35±0.22 | 95.38±0.06 | 71.21±0.20 | 79.51±0.05 |
| AFGRL | 77.62±0.49 | 89.88±0.33 | 93.27±0.17 | 95.69±0.10 | 71.05±0.32 | 79.20±0.17 |
| GraphECL | **80.17±0.15** | 89.91±0.35 | **93.51±0.12** | **95.81±0.12** | **71.75±0.22** | **82.69±0.33** |

Table 10: Graph classification results (%) on MUTAG and PROTEINS.

| Method | Graph-MLP | VGAE | CCA-SSG | DSSL | GraphCL | GraphECL |
|---|---|---|---|---|---|---|
| MUTAG | 75.8±2.0 | 84.4±0.6 | 85.8±0.4 | 87.2±1.5 | 86.8±1.3 | **88.5±1.2** |
| PROTEINS | 71.1±1.5 | 74.0±0.5 | 73.1±0.6 | 73.5±0.7 | 74.4±0.5 | **75.2±0.3** |

Table 11: The effect of hidden layers on GraphECL.

| Layers | Cora | Citeseer | Photo | Texas | Cornell | Wisconsin |
|---|---|---|---|---|---|---|
| 1 | 83.57±0.03 | 73.15±0.41 | 93.47±0.15 | 71.19±2.58 | 67.28±4.35 | 76.54±1.28 |
| 2 | **84.25±0.05** | **73.56±0.50** | **94.22±0.11** | 75.95±5.33 | 69.19±6.86 | **79.41±2.19** |
| 4 | 84.17±0.03 | 73.21±0.30 | 94.10±0.12 | **76.21±3.95** | **69.33±5.51** | 79.27±1.75 |

Table 12: The performance on the long-range graph benchmark PascalVOC-SP (Dwivedi et al., 2022)

| Method | BGRL | CCA-SSG | GraphECL |
|---|---|---|---|
| PascalVOC-SP | 0.1356±0.0087 | 0.1437±0.0095 | **0.1588±0.0091** |

pairs are more easily distinguishable from one-hop and two-hop neighbors based on representation similarity for homophilic graphs. This demonstration underscores that our GraphECL model effectively captures the semantic meaning of nodes, encouraging the separation of semantically dissimilar nodes.

Furthermore, we observe that the two-hop similarities in heterophilic graphs are significantly larger than those in homophilic graphs. This observation provides an explanation for GraphECL's strong performance, as it effectively captures the 1-hop structural information not only emphasizes homophily.

