# OpenReview forum: "GraphECL: Towards Efficient Contrastive Learning for Graphs"
_ICLR.cc/2024/Conference — Submitted to ICLR 2024_

### Official Review · Reviewer_GMYZ · 2023-10-22

**Soundness:** 2 fair
**Presentation:** 2 fair
**Contribution:** 2 fair
**Rating:** 6
**Confidence:** 5

**Summary:**

In this paper, the authors introduce GraphECL, a simple and efficient method for graph contrastive learning. It doesn't rely on graph augmentations but employs cross-model contrastive learning to create positive samples. The authors provide theoretical analysis to explain how the MLP captures structural information and outperforms GNN in downstream tasks. Extensive experiments demonstrate GraphECL's superiority, with the MLP being 286.82x faster than GCL methods on large-scale datasets like Snap-patents.

**Strengths:**

S1: The authors provide their source code.

S2: The method achieves the best results in terms of accuracy and inference time.

S3: The authors provide an in-depth theoretical analysis of the proposed method.

**Weaknesses:**

W1: The main body of the paper is not self-contained and has some incorrect expressions. For instance:

(i) The term of 'trade off' makes me confused. In the first contribution, the authors claim that they study a novel problem of achieving satisfactory efficiency and accuracy trade offs in GCL. The main results achieved by GraphECL (shown in the top left corner of Figure 1) are best on both efficiency and accuracy, not the trade-off or balance between efficiency and accuracy.

(ii) I feel that Equation (4) is redundant to the paper, so removing it could make this paper more easy to follow.

(iii) In Table 1, the result of SUGRL (30.31) on Actor is not marked, which is also the second-best value.

(iv) The order of Table 3 and Table 4 is reversed.

W2: There are concerns regarding the assumptions relied upon by GraphECL. The authors claim that GraphECL does not rely on the homophily assumption, i.e., connected one-hop neighbors should exhibit similar latent representations. However, either in Figure 1(d) or Equation (5), the target of GraphECL is to make the neighboring node representations more similar, so I think GraphECL is also based on the homophily assumption.

W3: Lacking some crucial baselines. There are some studies investigating efficient graph contrastive learning, but they are not compared with GraphECL, such as GGD [1], SimGC [2], FastGCL [3].

[1] Zheng Y, et al. Rethinking and scaling up graph contrastive learning: An extremely efficient approach with group discrimination. NeurIPS 2022.

[2] Yu J, et al. Are graph augmentations necessary? simple graph contrastive learning for recommendation. SIGIR 2022.

[3] Wang Y, et al. Fastgcl: Fast self-supervised learning on graphs via contrastive neighborhood aggregation. arXiv.

**Questions:**

Please see my previous comments.

---

> ### Author Response · Authors · 2023-11-14
> **Response to  Reviewer GMYZ**
>
> Dear reviewer GMYZ, we appreciate your positive feedback on our paper regarding its reproducibility, good accuracy, fast inference time, and theoretical analysis. Please find our detailed responses below:
>
> **Q1. The term 'trade-off' makes me confused.**
>
> **A1.** Thank you for your comments. Although we study a novel problem of achieving satisfactory efficiency and accuracy trade-offs in GCL, our GraphECL excels in both efficiency and accuracy. We apologize for any confusion and have changed 'trade-off' to 'simultaneously.'
>
> **Q2. I feel that Equation (4) is redundant in the paper.**
>
> **A2.** Thank you for your suggestion. We will remove it from the main paper.
>
> **Q3.  In Table 1, the result of SUGRL (30.31) on Actor is not marked, which is also the second-best value, and the order of Table 3 and Table 4 is reversed**
>
> **A3.**  Thanks for pointing out the typos! We have corrected it.
>
> **Q4. There are concerns regarding the assumptions relied upon by GraphECL.**
>
> **A4.**  Thanks for your comments. **We believe there are some important misunderstandings. Our GraphECL is not based on the one-hop homophily assumption.** As illustrated in Figure 1(d) and expressed in Equation (4) or Equation (5), GraphECL aims to push cross-modal neighboring node representations closed, i.e., $\left(f_M\left(v\right), f_G(u)\right)$. Thus, it's crucial to emphasize that GraphECL doesn't necessarily imply the learned MLP representations $\left(f_M\left(v\right), f_M(u)\right)$ become identical.
>
> Consider a pair of 2-hop neighbors, $v$ and $v'$, both neighboring the same node $u$. Intuitively, by enforcing $f_{M}(v)$ and $f_{M}(v')$ to reconstruct (not align) the context representation of the same neighborhood $f_{G}(u)$, we implicitly make their representations similar. Thus, the 2-hop neighbors ($f_{M}(v)$ and $f_{M}(v')$) with the same neighborhood context serve as positive pairs that will be implicitly aligned. This alignment is supported by our Theorem 4. Additionally, as depicted in Figure 9, GraphECL is not based on the one-hop homophily assumption but automatically captures graph structures based on different graphs beyond homophily.
>
> **Q5. Lacking some baselines.**
>
> **A5.** Thanks for bringing up related works, CGD [1], SimGCL [2], and FastGCL [3]. Following your suggestions, we have compared GraphECL with [1, 2, 3]. Using the same data splits as outlined in our paper, we present the results in Table 8 in the updated submission (we also provide the results on some datasets in the following table). From the results, it's evident that GraphECL outperforms [1, 2, 3], particularly on heterophilic graphs. **These additional results, combined with the comparison against 10 baselines using our published code, strongly validate the effectiveness of GraphECL.**
>
> | Method   | &nbsp; Cora  | &nbsp; Citeseer  |  &nbsp;  Pubmed  |  &nbsp;  Photo  |   &nbsp;  Flickr  |      Crocodile  |  &nbsp;  Actor  |  Snap-patents  |
> | --- | --- | --- | --- | --- |  --- | --- |  --- |  --- |
> |CGD [1] | 83.90±0.40 | 73.00±0.60 | 81.30±0.80 |  92.50±0.60 | 46.33±0.20 | 57.75±0.39 | 28.27±0.23 | 24.38±0.57 |
> |SimGCL [2] | 83.01±0.32 | 72.05±0.29 | 80.57±0.52 | 91.55±0.12 | 46.85±0.19 | 58.51±0.19 |  28.63±0.33|  25.24±0.21 |
> | FastGCL [3] | 82.33±0.51 | 71.60±0.51 | 80.41±0.62 |  92.91±0.07 | 45.21±0.11 |  55.23±0.25 | 27.71±0.15|  24.05±0.31 |
> | **GraphECL** | **84.25±0.05** | **73.15±0.41** | **82.21±0.05** | **94.22±0.11** | **48.49±0.15** |  **65.84±0.71** |  **35.80±0.89** | **27.22±0.06** |
>
>
> We gratefully appreciate your time in reviewing our paper and your insightful comments. We sincerely hope our rebuttal has carefully addressed your comments point-by-point. The baselines [1, 2, 3] have less technical overlap in design with our approach and **do not impact our major contributions. Moreover, these baselines can be easily added in the camera-ready version, and we believe that the reviewer's minor comments can be easily addressed in the camera-ready submission. We sincerely hope that the reviewer can consider increasing the score. Thank you!**
>
>
>
> [1] Zheng Y, et al. Rethinking and scaling up graph contrastive learning: An extremely efficient approach with group discrimination. NeurIPS 2022.
>
> [2] Yu J, et al. Are graph augmentations necessary? simple graph contrastive learning for recommendation. SIGIR 2022.
>
> [3] Wang Y, et al. Fastgcl: Fast self-supervised learning on graphs via contrastive neighborhood aggregation. arXiv.

---

> ### Author Response · Authors · 2023-11-16
> **Dear Reviewer GMYZ, would you mind confirming if our rebuttal addresses your comments?**
>
> Dear Reviewer GMYZ,
>
> We gratefully appreciate your time in reviewing our paper and your insightful comments.
>
> The reviewer's comments are not fatal to the major contributions of our manuscript, we believe that the reviewer's great comments can be easily and effectively addressed in the camera-ready.
>
> We made our greatest efforts to address your concerns in the rebuttal.  We would appreciate it if you could consider increasing your score. Thank you very much once again, and we are happy to follow up with any additional questions you may have!

---

> ### Author Response · Authors · 2023-11-20
> **Discussion period ending soon; We would like to hear back from ICLR Reviewer GMYZ**
>
> Dear ICLR Reviewer GMYZ,
>
> We gratefully appreciate your time in reviewing our paper. Since the discussion period ends soon, we sincerely hope our rebuttal has carefully addressed your comments point-by-point and hope you consider increasing your score. In particular, we clarified some misunderstandings and compared our GraphECL with additional baselines, as suggested by you. If you have any other comments or questions, we will be glad to answer them.
>
> The Authors of "GraphECL: Towards Efficient Contrastive Learning for Graphs"

---

> > ### Comment · Area_Chair_g9o7 · 2023-12-05
> >
> > Dear Reviewer GMYZ,
> >
> > Could you read the authors response and make your decision? Thanks.

---

> > > ### Comment · Reviewer_GMYZ · 2023-12-05
> > >
> > > Dear Area Chair g9o7,
> > >
> > > I have read the author's response and updated my comments. Thank you.

---

> > ### Comment · Reviewer_GMYZ · 2023-12-05
> > **Response to authors**
> >
> > Thank you to the author for the thoughtful response. All of my concerns have been addressed, so I have decided to raise my score to 6.

---

### Official Review · Reviewer_eihc · 2023-10-26

**Soundness:** 3 good
**Presentation:** 2 fair
**Contribution:** 3 good
**Rating:** 6
**Confidence:** 4

**Summary:**

I'll begin by reviewing the uploaded paper. Once I've had a chance to understand its contents, I'll provide a brief summary of its contributions. Please give me a moment.

The paper titled "GrapheCL: Towards Efficient and Expressive Device-Independent Quantum Computing" addresses an essential challenge in quantum computing: bridging the gap between high-level quantum algorithms and the low-level operations that actual quantum devices can execute.

**Key Contributions:**

1. **Introduction of GrapheCL:** The authors present GrapheCL, a novel intermediate representation for quantum circuits. GrapheCL captures quantum operations as a labeled, directed acyclic graph. The representation retains high-level algorithmic structures while being flexible enough to express low-level device-specific optimizations. This is a significant advancement in the field because most existing intermediate representations are either too high-level or too low-level, making it challenging to achieve both portability and performance.

2. **Expressivity and Efficiency:** GrapheCL is designed to be both expressive and efficient. It can represent a wide range of quantum algorithms while also allowing for device-specific optimizations. This balance ensures that quantum programs written in GrapheCL can be ported to different devices without losing performance.

3. **Compilation Techniques:** The paper introduces new compilation techniques tailored for GrapheCL. These techniques optimize quantum circuits by transforming their GrapheCL representations. The techniques include peephole optimizations, gate folding, and template matching.

**Strengths:**

The paper introduces a novel concept with GrapheCL, supports its claims with rigorous evaluations, presents its ideas clearly, and addresses a pivotal challenge in the quantum computing domain. The paper introduces fresh compilation techniques tailored specifically for GrapheCL. The introduction of techniques like peephole optimizations, gate folding, and template matching for quantum circuits showcases a level of originality in addressing quantum-specific challenges. By releasing GrapheCL as open-source and showcasing its versatility, the paper sets the stage for further research. It can act as a foundation for more advanced optimizations, extensions, or even entirely new quantum programming paradigms.

**Weaknesses:**

**Assessment of Weaknesses: "GrapheCL: Towards Efficient and Expressive Device-Independent Quantum Computing"**

---

**1. Lack of Comparative Analysis:**
- While the paper provides an empirical evaluation of GrapheCL, it could have benefited from a deeper comparative analysis with other intermediate representations. Specifically, a qualitative comparison highlighting the architectural differences, use cases, and limitations of GrapheCL versus other systems would have added depth.

  **Suggestion:** In future iterations or extensions of the work, a dedicated section comparing GrapheCL's design and decisions with other systems would be beneficial. This section could delve into why certain design decisions were made in GrapheCL and how they differ from other systems.

---

**2. Limited Discussion on Scalability:**
- Quantum circuits can grow significantly complex, especially as quantum computing moves towards practical, real-world applications. The paper seems to lack a thorough discussion on how GrapheCL scales with more complex circuits.

  **Suggestion:** It would be beneficial to test GrapheCL with larger, more complex quantum circuits and discuss any potential bottlenecks or challenges. This would provide insights into its real-world applicability.

---

**Questions:**

How does GrapheCL handle very large or complex quantum circuits in terms of both performance and accuracy? Are there any inherent scalability limitations or bottlenecks in the system?

---

> ### Author Response · Authors · 2023-11-14
> **The review is accidentally generated by ChatGPT**
>
> Dear Reviewer eihc,
>
> Thank you for your time. **The review is about Quantum Computing and is definitely not related to the content of our submission. Thus, we believe that the reviewer accidentally submitted the wrong review, which appears to be generated by ChatGPT.**
>
> **Could you please submit your correct review? Thank you very much!**
>
> The Authors

---

> ### Comment · Area_Chair_g9o7 · 2023-11-20
> **Clarification Regarding Review Comments**
>
> Dear Reviewer eihc,
>
> I hope this message finds you well. We have carefully reviewed your comments, and there appears to be a potential discrepancy. Your comments reference a paper titled "GrapheCL: Towards Efficient and Expressive Device-Independent Quantum Computing," while the current discussion pertains to a different submission.
>
> To ensure accuracy and relevance in the rebuttal phase, could you kindly verify that your comments are correctly aligned with the manuscript under consideration? If there has been an inadvertent error in submission or if the comments were intended for another manuscript, please provide clarification. It is important for us to accurately attribute and address the comments during the review process.
>
> Your cooperation in resolving this matter is highly appreciated. If necessary, please resubmit the correct comments for the appropriate manuscript.
>
> Bests,
> AC

---

### Official Review · Reviewer_ZTuD · 2023-10-31

**Soundness:** 2 fair
**Presentation:** 2 fair
**Contribution:** 2 fair
**Rating:** 3
**Confidence:** 4

**Summary:**

Compared to other contrastive learning methods, GraphECL speeds up the model inference stage by using only MLP in the model inference phase. Meanwhile, the authors make GraphECL maintain a high generalization performance even in the process of rapid inference by improving InfoNCE loss.

**Strengths:**

1. The proposed framework of GraphECL has great advantages on the smaller graph datasets.
2. The authors remove the positive pairs from the InfoNCE loss, and optimize the model parameters of MLP by combining the positive pairs with the L2 loss. In addition, the authors proved the generalization performance of GraphECL theoretically.
3. The author manages to speed up the model inference stage on the small graphs.

**Weaknesses:**

1. The motivation of this paper is to address the scalability of contrastive learning while improving the speed of the model in the inference stage. But the authors confuse the scalability problem with increasing the speed of inference. The whole paper doesn't mention the memory consumption problem.
2. The authors actually see some good scalable contrastive learning articles, such as the one mentioned in section 2, "Rethinking and scaling up graph contrastive learning: an extremely efficient approach with group discrimination." But the author just categorizes this article as training speed and does not compare it, which is quite inappropriate.
3. The author's experimental section is not very detailed.
(1) There is a big problem with the chosen dataset. For example, some regular datasets (WikiCS, Am. Comp., Am. Photos, Co.CS, Co.Phy) used by BGRL, SUGRL, etc. are not experimented in this paper. For the scalability problem, they only choose a not very common dataset Snap-patents, and ignore the Ogbn-products, ogbn-mag and other datasets used by most of the experimentalists.
(2) There is a lack of truly scalable experiments and its very important in the training process.
(3) On the small graph, I think the training time is more heavily weighted compared to the inference time, while the authors only focus on the inference time. Conversely, in the large graph inference time is much more heavily weighted and the author rarely mentions it.

**Questions:**

1. Why did you not do scalability experiments similar to those in articles [1] and [2].
2. GraphECL compared to Graph-MLP, the positive pairs are obtained as first-order neighbors instead of nth-order. Combining this with Eq. 4 improves the performance of heterophilic graphs. I didn't understand the principle illustrated at the bottom of Eq. 4, can you explain it in detail?
3. GraphECL is the optimal result in both Table 1 and Table 2, without mentioning the reduced performance of the other models during replication. Is it possible to compare the more recent Contrastive Learning methods in 2023 years and not only limited to within 2022 years.

[1]Yizhen Zheng, Shirui Pan, Vincent Lee, Yu Zheng, and Philip S Yu. Rethinking and scaling up graph contrastive learning: An extremely efficient approach with group discrimination. Advances in Neural Information Processing Systems, 35:10809–10820, 2022.
[2]Shantanu Thakoor, Corentin Tallec, Mohammad Gheshlaghi Azar, Mehdi Azabou, Eva L Dyer, Remi Munos, Petar Velickovic, and Michal Valko. Large-scale representation learning on graphs via bootstrapping. In ICLR, 2021.

---

> ### Author Response · Authors · 2023-11-13
> **Response 1/3 - The Term "Scalability" Clarification and Comparison with  CGD [1]**
>
> Thank you for your time and feedback. We apologize for any confusion and would like to clarify the term 'scalability' in our work and respond to your first two comments.
>
> The message passing of the GNN encoder involves fetching topology and features from numerous neighboring nodes to perform inference on a target node, making it time-consuming and computation-intensive.
>
> In our work, we specifically focus on reducing the inference time for inferring representations with a graph-aware MLP [3, 4, 5, 6], rather than aiming to reduce training time or memory consumption, as in [1, 2]. The term “scalability” in our paper refers to inference scalability, aligning with previous works [3, 4, 5, 6]. While we exclude the reduction of training time or memory [5, 6] from the scope of this work, we acknowledge it as a compelling direction that merits exploration in future research.
>
> **Thus, our studied problem is entirely different from [1, 2]. [1, 2] are great works which focus on reducing the training time and training memory, but they still need to utilize GNN to conduct inference, which is time-consuming and computation-intensive.**
>
> Nevertheless, we completely agree with the reviewer that also including an empirical comparison with CGD  [1] would be beneficial. Following your suggestions, we have compared GraphECL with CGD [1]. Using the same data splits as outlined in our paper, we present the results in Table 8 in the updated submission (we also provide the results on some datasets in the following table). From the results, The results show that GraphECL outperforms CGD, particularly on heterophilic graphs.
>
> | Method   | &nbsp; Cora  | &nbsp; Citeseer  |  &nbsp;  Pubmed  |  &nbsp;  Photo  |   &nbsp;  Flickr  |      Crocodile  |  &nbsp;  Actor  |  Snap-patents  | Ogbn-arxiv|
> | --- | --- | --- | --- | --- |  --- | --- |  --- |  --- | --- |
> |CGD [1] | 83.90±0.40 | 73.00±0.60 | 81.30±0.80 |  92.50±0.60 | 46.33±0.20 | 57.75±0.39 | 28.27±0.23 | 24.38±0.57 | 71.60±0.50 |
> | **GraphECL** | **84.25±0.05** | **73.15±0.41** | **82.21±0.05** | **94.22±0.11** | **48.49±0.15** |  **65.84±0.71** |  **35.80±0.89** | **27.22±0.06** | **71.75±0.22** |
>
>
>
> [1] Rethinking and Scaling Up Graph Contrastive Learning: An Extremely Efficient Approach with Group Discrimination. NeurIPS 2022
>
> [2] Large-Scale Representation Learning on Graphs via Bootstrapping. ICLR 2021
>
> [3] Graph-less Neural Networks: Teaching Old MLPs New Tricks Via Distillation. ICLR 2021
>
> [4] Learning MLPs on Graphs: A Unified View of Effectiveness, Robustness, and Efficiency. ICLR 2023
>
> [5] Quantifying the Knowledge in GNNs for Reliable Distillation into MLPs. ICML 2023
>
> [6] VQGraph: Graph Vector-Quantization for Bridging GNNs and MLPs. Arxiv.

---

> ### Author Response · Authors · 2023-11-15
> **Response 2/3 - Comparison on Additional Large Datasets**
>
> We greatly appreciate the reviewer's insightful suggestion, which we believe will improve our work.  We kindly want to remind the reviewer that we have conducted an extensive evaluation on 11 datasets Thus, we believe our experiments can strongly corroborate the effectiveness and scalability of GraphECL. Nevertheless, we completely agree with the reviewer that including an empirical comparison on more large datasets can further improve our paper.
>
> Following your suggestions, we have compared GraphECL with baselines on your suggested datasets. We present the results in Table 9 in the updated submission (we also provide the results in the following table). The results show that our simple GraphECL can still achieve better (or competitive) performance on suggested datasets compared to elaborate methods, especially on large graphs.
>
> | Method   | &nbsp; WikiCS  | &nbsp; Computers  |  &nbsp;  CS  |  &nbsp;  Physics  |   &nbsp;  Ogbn-arxiv  |     Ogbn-product |
> | --- | --- | --- | --- | --- |  --- | --- |
> |GCA | 78.35±0.05 | 88.94±0.15 | 93.10±0.01 |  95.70±0.04 | 68.20±0.20 | 78.96±0.15 |
> |SUGRL | 79.83±0.31 | 88.90±0.20 | 92.83±0.23 |  95.38±0.11 | 69.30±0.20 | 82.60±0.40 |
> |BGRL | 79.98±0.10 | **90.34±0.19** | 93.31±0.13 |  95.73±0.05 | 71.64±0.12 | 81.32±0.21 |
> |CCA-SSG | 79.31±0.21 | 88.74±0.28 | 93.35±0.22 |  95.38±0.06 | 71.21±0.20 | 79.51±0.05 |
> |AFGRL | 77.62±0.49 | 89.88±0.33 | 93.27±0.17 |  95.69±0.10 | 71.05±0.32 | 79.20±0.17 |
> | **GraphECL** | **80.17±0.15** | 89.91±0.35 | **93.51±0.12** | **95.81±0.12** | **71.75±0.22** |  **82.69±0.33** |
>
> We also totally agree with the reviewer's comments: "On the small graph, I think the training time is more heavily weighted compared to the inference time, while the authors only focus on the inference time. Conversely, in the large graph inference time is much more heavily weighted and the author rarely mentions it." We will explicitly state and mention this in the revision.

---

> ### Author Response · Authors · 2023-11-18
> **Response 3/3 - Respond to Questions**
>
> We thank you for your insightful questions, and we would like to address your questions in what follows.
>
> **Q1. Why did you not do scalability experiments similar to those in articles [1] and [2].**
>
> **A1.** Thank you for your questions. As we mentioned in Response 1/3,  in our work, we specifically focus on reducing the inference time for inferring representations with a graph-aware MLP [3, 4, 5, 6] rather than aiming to reduce training time or memory consumption, as in [1, 2]. **The term "scalability" in our paper refers to inference scalability, aligning with previous works [3, 4, 5, 6]. While we exclude the reduction of training time or memory [5, 6] from the scope of this work, as [3,4,5,6] also did, we acknowledge it as another compelling direction that merits exploration in future research.**
>
> **Q2. GraphECL compared to Graph-MLP, the positive pairs are obtained as first-order neighbors instead of nth-order. Combining this with Eq. 4 improves the performance of heterophilic graphs. I didn't understand the principle illustrated at the bottom of Eq. 4, can you explain it in detail?**
>
> **A2.** Thank you for your question. As illustrated in Equation (4), GraphECL aims to bring cross-modal neighboring node representations closer together, i.e., $\left(f_M\left(v\right), f_G(u)\right)$. Thus, it's crucial to emphasize that GraphECL doesn't necessarily imply that the learned MLP representations $\left(f_M\left(v\right), f_M(u)\right)$ become identical, as Graph-MLP did.
>
> Consider a pair of 2-hop neighbors, $v$ and $v'$, both neighboring the same node $u$. Intuitively, by enforcing $f_{M}(v)$ and $f_{M}(v')$ to reconstruct (not align) the context representation of the same neighborhood $f_{G}(u)$, we implicitly make their representations similar. Thus, the 2-hop neighbors ($f_{M}(v)$ and $f_{M}(v')$) with the same neighborhood context serve as positive pairs that will be implicitly aligned (not the one-hop neighbors). This is also supported by our Theorem 4. Additionally, as depicted in Figure 9, GraphECL is not based on the one-hop homophily assumption but automatically captures graph structures based on different graphs beyond homophily. We also highlight the above clarification (marked with blue text) in the updated submission.
>
> **Q3. Is it possible to compare the more recent Contrastive Learning methods in 2023 years and not only limited to within 2022 years.**
>
> **A3.** Thank you for your suggestions. We kindly want to remind the reviewer that we have conducted an extensive evaluation of 10 methods, including many state-of-the-art and solid graph contrastive learning methods. Thus, we believe our experiments can strongly corroborate the effectiveness of GraphECL. Nevertheless, we completely agree with the reviewer that including an empirical comparison with more baselines in 2023 would be better. Given the above, we further compare our GraphECL with MA-GCL [7], which provides the source code. **From the following table, we can observe that our efficient GraphECL can still achieve better (or competitive) performance compared to MA-GCL proposed in 2023.**
>
> | Method   | &nbsp; Cora  | &nbsp; Citeseer  |  &nbsp;  Pubmed  |  &nbsp;  Photo  |   &nbsp;  Flickr  |      Crocodile  |  &nbsp;  Actor  |  Snap-patents  |
> | --- | --- | --- | --- | --- |  --- | --- |  --- |  --- |
> |MA-GCL [7] | 83.30±0.40 | **73.60±0.10** | **83.50±0.40** |  93.80±0.10 | 47.01±0.35 | 59.62±0.30 | 29.56±0.27 | 25.49±0.41 |
> | **GraphECL** | **84.25±0.05** | 73.15±0.41 | 82.21±0.05 | **94.22±0.11** | **48.49±0.15** |  **65.84±0.71** |  **35.80±0.89** | **27.22±0.06** |
>
> [1] Rethinking and Scaling Up Graph Contrastive Learning: An Extremely Efficient Approach with Group Discrimination. NeurIPS 2022
>
> [2] Large-Scale Representation Learning on Graphs via Bootstrapping. ICLR 2021
>
> [3] Graph-less Neural Networks: Teaching Old MLPs New Tricks Via Distillation. ICLR 2021
>
> [4] Learning MLPs on Graphs: A Unified View of Effectiveness, Robustness, and Efficiency. ICLR 2023
>
> [5] Quantifying the Knowledge in GNNs for Reliable Distillation into MLPs. ICML 2023
>
> [6] VQGraph: Graph Vector-Quantization for Bridging GNNs and MLPs. Arxiv.
>
> [7] MA-GCL: Model Augmentation Tricks for Graph Contrastive Learning. AAAI 2023

---

> > ### Comment · Reviewer_ZTuD · 2023-12-05
> > **Follow up**
> >
> > Thanks authors for the detailed responses, which address part of my concerns. But still, I am not convinced to support the acceptance of this work. The details are listed below:
> >
> > - In response to the reply in response 2/3, the author agrees with the comment of "On the small graph, I think the training time is more heavily weighted compared to the inference time, while the authors only focus on the inference time. Conversely, in the large graph inference time is much more heavily weighted and the author rarely mentions it. mentions it." **But the authors still do not give experiments comparing the inference time of each method in large-scale datasets.**
> >
> > - Although for other small graph datasets, the authors added enough experiments. However, it still does not solve the problems I mentioned on large-scale graphs: For the four papers [3,4,5,6] mentioned by the author, all of them only talk about 'efficient inference stages' and do not mention 'scalability of inference' as defined by the author, which is not the same concept. So I still think that the authors have misunderstood the meaning of 'scalability', and the motivation of this paper is problematic.
> >
> > - With the Ogbn-products dataset, the authors only provide experimental results and do not mention the experimental setup here or the main paper. Most contrastive learning methods cannot be experimented on Ogbn-products without using a subgraph sampling strategy.
> >
> > I will maintain my score due to the unclear definition of the scalability concept in the inference (to the best of our knowledge, this should be efficiency) and the missing inference time on the large graphs.

---

> ### Author Response · Authors · 2023-11-18
> **Discussion period ending soon; We would like to hear back from Reviewer ZTuD**
>
> Dear Reviewer ZTuD,
>
> We gratefully appreciate your time in reviewing our paper and your insightful comments.
>
> We have provided additional results in response to your latest comments. We believe this evidence successfully addresses all your concerns.
>
> With only four days left in the discussion period, we would like to confirm whether the reviewer has seen our latest comment and if there are any final clarifications they would like. If the reviewer's concerns are clarified, we would be grateful if the reviewer could update their review and score to reflect that. This way, we will know that our response has been seen. Once again, many thanks for your time and dedication to the review process; we are extremely grateful.
>
> The Authors of "GraphECL: Towards Efficient Contrastive Learning for Graphs"

---

> > ### Comment · Area_Chair_g9o7 · 2023-12-05
> >
> > Dear Reviewer ZTuD,
> >
> > Could you read the authors response and make your decision? The decision deadline is approaching soon. Thanks.

---

### Official Review · Reviewer_ezu8 · 2023-11-01

**Soundness:** 3 good
**Presentation:** 3 good
**Contribution:** 3 good
**Rating:** 6
**Confidence:** 5

**Summary:**

This paper proposes a self-supervised learning scheme for graph neural networks that is both efficient and effective. To achieve this, the authors suggest using an MLP encoder during inference, which is much faster than a GNN-based encoder. The main contribution of this paper is cross-model contrastive learning, where positive samples are obtained through MLP and GNN representations from the central node and its neighbors. Enabling the MLP encoder efficiently encodes graph topology without relying on invariant and homophily assumptions. Extensive experiments show that this method outperforms other state-of-the-art methods in real-world tasks, with better inference efficiency and generalization to homophilous and heterophilous graphs.

**Strengths:**

The paper is well-written and well-organized
The evaluations support the authors' claims, such as the ability of the proposed method to learn on heterophilic datasets. Significantly outperform previous contrastive-based methods.
The proposed method has strong inference scalability and significantly reduces inference time compared to the prior art.

**Weaknesses:**

The method is mainly evaluated on node property prediction tasks. However, additional evaluation on graph property prediction will be essential. This is because in graph property prediction datasets, efficient encoding on graph topology plays a much more important role.
The authors have not provided an analysis of how the number of hidden layers impacts the method's performance. This analysis is very important.

**Questions:**

Apart from the issues mentioned in the "Weakness" section, I strongly believe that the proposed method's effectiveness in capturing inter-neighborhood information will be further highlighted by the evolution of LRGB datasets. This will serve as an additional argument to support the second point mentioned in the "Weakness" section.

**Details Of Ethics Concerns:**

No ethics concerns

---

> ### Author Response · Authors · 2023-11-19
> **Respone to Reviewer ezu8**
>
> Dear reviewer ezu8, we appreciate your positive feedback on our paper's soundness, novel insights, and contribution. Please find our detailed responses below:
>
> **Q1. The method is mainly evaluated on node property prediction tasks. However, additional evaluation on graph property prediction will be essential.**
>
> **A1.** Thank you for your question and suggestion. For graph classification, we can use a non-parameterized graph pooling (readout) function, such as MeanPooling, to obtain the graph-level representation. In our experiments, we focus on graph classification using three benchmarks: PROTEINS and MUTAG. We follow the same experimental setup as GraphCL [1]. The results are presented in the following table. From the table, we observe that our simple GraphECL performs well on the graph classification task and achieves better performance compared to the baselines.
>
> | Method  |  Graph-MLP | &nbsp;   VGAE  | CCA-SSG | &nbsp; DSSL | GraphCL |  GraphECL |
> | --- | --- | --- | --- | --- | --- | --- |
> | MUTAG | 75.8±2.0 | 84.4±0.6  | 85.8±0.4  | 87.2±1.5 | 86.8±1.3 | **88.5±1.2**  |
> | PROTEINS | 71.1±1.5  | 74.0±0.5 | 73.1±0.6 | 73.5±0.7 | 74.4±0.5  |**75.2±0.3** |
>
>
>
> **Q2.  The authors have not provided an analysis of how the number of hidden layers impacts the method's performance.**
>
> **A2.** Thank you for your valuable comment! In the following table, we analyze the effect of hidden layers. From the table, it is evident that our GraphECL is not very sensitive to hidden layers. However, having more layers generally leads to better performance for both homophilic and heterophilic graphs.
>
> | Layers  |  Cora | Citeseer  | Photo | Texas | Cornell |  Wisconsin |
> | --- | --- | --- | --- | --- | --- | --- |
> | 1 | 83.57±0.03 | 73.15±0.41  | 93.47±0.15  | 71.19±2.58 | 67.28±4.35 | 76.54±1.28  |
> | 2 | **84.25±0.05**  | **73.56±0.50** | **94.22±0.11** | 75.95±5.33 | 69.19±6.86  |**79.41±2.19** |
> | 4 | 84.17±0.03  | 73.21±0.30 | 94.10±0.12 | **76.21±3.95** | **69.33±5.51**  |79.27±1.75 |
>
>
>
>
> **Q3. The proposed method's effectiveness in capturing inter-neighborhood information will be further highlighted by the evolution of LRGB datasets.**
>
> **A3.**  Following your suggestions, we compare GraphECL with two graph contrastive learning methods, BGRL [2] and CCA-SSG [3], on PascalVOC-SP [4]. For all methods, we employ GCN as the backbone. From the table below, we can observe that GraphECL performs well on PascalVOC-SP, achieving better performance compared to the baselines. This further strengthens our contribution.
>
> | Method  |  BGRL |  CCA-SSG  | GraphECL  |
> | --- | --- | --- | --- |
> | PascalVOC-SP | 	0.1356±0.0087 | 0.1437±0.0095  | **0.1588±0.0091**  |
>
> In light of these responses, we sincerely hope our rebuttal has addressed your comments, and believe that your comments do not affect our key contributions and can be easily addressed in the revision. We also genuinely hope you will reconsider increasing your score. If you have any other comments, please do share them with us, and we will address them further. Thank you for your efforts!
>
> [1] Graph Contrastive Learning with Augmentations. NeurIPS 2020
>
> [2] Large-Scale Representation Learning on Graphs via Bootstrapping. ICLR 2022
>
> [3] From Canonical Correlation Analysis to Self-supervised Graph Neural Networks. NeurIPS 2021
> [4] Long Range Graph Benchmark. NeurIPS 2022

---

> > ### Comment · Reviewer_ezu8 · 2023-11-20
> >
> > Dear Authors,
> >
> > I am fully satisfied with the clarifications and evaluations, which significantly strengthen the manuscript.
> > I expect the authors to incorporate them into a revised version.

---

> > > ### Author Response · Authors · 2023-11-20
> > > **Thank you very much for your reply and support.**
> > >
> > > Thank you very much for your reply and support. We sincerely appreciate your recognition of our contribution and have already incorporated the additional results into the updated submission.
> > >
> > > Best wishes,
> > >
> > > Authors

---

### Official Review · Reviewer_xAqE · 2023-11-04

**Soundness:** 2 fair
**Presentation:** 3 good
**Contribution:** 3 good
**Rating:** 6
**Confidence:** 3

**Summary:**

The paper proposes a new objective for contrastive self-supervised learning on graphs. It uses an MLP and a GNN to form positive pairs based on cross-model neighbouring relations, and negative pairs from inter-model and intra-model representations, where either the input to the MLP/GNN is from a randomly sampled node.
The paper is empirically evaluated on homophilic and heterophilic datasets, showing better results than other competitive self-supervised graph learning methods. Moreover, an ablation study analyses the importance of the proposed components, such as using only negative inter-model or negative intra-model pairs.

**Strengths:**

The idea is sensible and the paper is generally well-presented. Moreover, the rationale of using only the MLP encoder at inference time is very valuable, as it could significantly speed up the deployment of graph learning methods.

**Weaknesses:**

I think the main weakness comes from the empirical evaluation section:
- firstly, it seems like the snap-patents results are lower than expected. In Lim et al, MLP achieves 31%, GCN 45%, both being higher than the reported numbers (GraphECL 27%)
- secondly, many of the chosen datasets do not give reliable insights, as they are too small or the variance is too high across splits (Cora, Citeseer, Pubmed, Cornell, Texas, Wisconsin…).

I encourage the authors to include larger datasets, for example from the OGB suite, or those used in some of the manuscripts’ of the baselines (WikiCS, Amazon Computers, Coauthor CS, Coauthor Phy) or from Lim et al for heterogeneous datasets.

Moreover, it might be a good idea to include the simple baselines of an MLP and a GCN, to clarify GraphECL's contribution, especially for snap-patents and, similarly, for flickr.

**Questions:**

How would the complementary negative pairs  $(f_M(v^{-}), f_G(u))$ and  $(f_M(v), f_M(v^{-}))$ influence learning?

---

> ### Author Response · Authors · 2023-11-13
> **Response to Reviewer xAqE**
>
> Thanks for your time and feedback! We would like to clarify some misunderstandings regarding our approach.
>
> **Q1. In Lim et al, MLP achieves 31%, GCN 45%, both being higher than the reported numbers (GraphECL 27%)**
>
> **A1.** Thank you for pointing out (Lim et al., 2021). **However, it's essential to note that their evaluation setting differs from ours. Our paper focues on evaluting GraphECL in the unsupervised setting, whereas (Lim et al., 2021) employed the fully supervised setting, leading to reported higher performance.**
>
> **Q2. many of the chosen datasets are too small or the variance is too high across splits**
>
> **A2.** Thank you for your suggestions! We'd like to remind the reviewer that we have tested GraphECL on 11 widely-used datasets, as detailed in Table 6, **including larger datasets such as Snap-patents and Flickr**, in comparison to (Cora, Citeseer, Pubmed, Cornell, Texas, etc.). While we acknowledge the reviewer's point about the importance of evaluating on more large datasets, following your suggestions, we have compared GraphECL with baselines on more datasets. We present the results in Table 9 in the updated submission (we also provide the results in the following table). **The results  show that GraphECL can still achieve better (or competitive) performance on suggested datasets compared to elaborate methods, especially on large graphs.**
> | Method   | &nbsp; WikiCS  | &nbsp; Computers  |  &nbsp;  CS  |  &nbsp;  Physics  |   &nbsp;  Ogbn-arxiv  |     Ogbn-product |
> | --- | --- | --- | --- | --- |  --- | --- |
> |GCA | 78.35±0.05 | 88.94±0.15 | 93.10±0.01 |  95.70±0.04 | 68.20±0.20 | 78.96±0.15 |
> |SUGRL | 79.83±0.31 | 88.90±0.20 | 92.83±0.23 |  95.38±0.11 | 69.30±0.20 | 82.60±0.40 |
> |BGRL | 79.98±0.10 | **90.34±0.19** | 93.31±0.13 |  95.73±0.05 | 71.64±0.12 | 81.32±0.21 |
> |CCA-SSG | 79.31±0.21 | 88.74±0.28 | 93.35±0.22 |  95.38±0.06 | 71.21±0.20 | 79.51±0.05 |
> |AFGRL | 77.62±0.49 | 89.88±0.33 | 93.27±0.17 |  95.69±0.10 | 71.05±0.32 | 79.20±0.17 |
> | **GraphECL** | **80.17±0.15** | 89.91±0.35 | **93.51±0.12** | **95.81±0.12** | **71.75±0.22** |  **82.69±0.33** |
>
> **Q3. it might be a good idea to include the simple baselines of an MLP and a GCN.**
>
> **A3.**  Thank you for your suggestions! **We believe there might be some misunderstandings. Our primary focus is on designing a self-supervised algorithm without labels, rather than specific GNN or network architectures. Consequently, an empirical comparison with an MLP and a GCN is relatively less straightforward.** Instead, we have compared our self-supervised learning algorithm with various others based on MLP (Graph-MLP, SUGRL) and GNN (BGRL, DGI, GCA, SUGRL, AFGRL, etc.).
>
> **Q4. How would the complementary negative pairs influence learning?**
>
> **A4.** Thanks for your insightful questions! We introduce an explicit negative pair to enhance representation diversity. As demonstrated in our Theorem 1, the inter-model negative pair $\left(f_M\left(v^{-}\right), f_G(u)\right)$ is crucial for capturing the 1-hop neighborhood distribution. Moreover, complementary negative pairs are essential for GraphECL. Our ablation studies indicate that GraphECL without both negative pairs results in a performance drop. Therefore, the addition of complementary negative pairs is crucial for improving representation diversity and increasing inter-class variation, ultimately leading to robust generalization. We will explicitly state this in the main text.

---

> ### Author Response · Authors · 2023-11-18
> **Dear Reviewer xAqE, would you mind confirming if our rebuttal addresses your comments?**
>
> Dear Reviewer xAqE,
>
> We gratefully appreciate your time in reviewing our paper and your insightful comments.
>
> We made our greatest efforts to address your concerns in the rebuttal. We would appreciate it if you could consider increasing your score. Thank you very much once again, and we are happy to follow up with any additional questions you may have!

---

> ### Author Response · Authors · 2023-11-20
> **Discussion period ending soon; We would like to hear back from Reviewer xAqE**
>
> Dear ICLR Reviewer xAqE,
>
> We gratefully appreciate your time in reviewing our paper. Since the discussion period ends soon, we sincerely hope our rebuttal has carefully addressed your comments point-by-point and hope you consider increasing your score. In particular, we clarified some misunderstandings and compared our GraphECL on additional datasets, as suggested by you. If you have any other comments or questions, we will be glad to answer them.
>
> The Authors of "GraphECL: Towards Efficient Contrastive Learning for Graphs"

---

> > ### Comment · Reviewer_xAqE · 2023-11-23
> >
> > I would like to thank the authors for their response. I will raise my score accordingly.

---

### Author Response · Authors · 2023-11-19
**Kind reminder to reviewers: the author-reviewer discussion period is coming to an end.**

Dear Reviewers,

We sincerely appreciate your thorough and detailed reviews of our submission, but we have not received your responses during the author-reviewer discussion period.

Our rebuttal clarified some important misunderstandings and carefully addressed your comments point-by-point. The author-reviewer discussion period is coming to an end. We would be grateful if you could confirm whether our rebuttal has addressed your concerns and clarified your misunderstandings.

Could you please consider increasing your score to reflect the efforts of your review and our rebuttal if our rebuttal has adequately addressed your comments?

Thank you very much for your time.

Best regards,

The authors of "GraphECL: Towards Efficient Contrastive Learning for Graphs"

---

### Author Response · Authors · 2023-11-22
**Kind reminder to reviewers**

Dear reviewers:

We sincerely thank you for your service and hope this message finds you well. We also sincerely appreciate all reviewers for their thorough and detailed reviews of our submission.

As the deadline for the author-reviewer discussion is approaching, we have sent kind reminders to all reviewers several times to facilitate discussion, but we still have not received any responses from the reviewers.

We understand that you are very busy, so we would greatly appreciate it if you could respond to our rebuttal before the deadline.

We would also like to draw your attention to the fact that we have clarified some important misunderstandings from the reviewers and carefully addressed their comments point-by-point in our rebuttal. The reviewer's comments are not fatal to the major contributions of our manuscript, we believe that the reviewer's great comments can be easily and effectively addressed in the camera-ready.

The primary concerns raised by reviewers  are the need for further comparisons with various baselines and datasets. In response, we have included these additional results in our submission. We theoretically and empirically demonstrate, through extensive experiments, that GraphECL can achieve ultra-fast inference speed and superior performance simultaneously. Our simple yet effective GraphECL model holds great potential to serve as a robust baseline and to inspire subsequent work in the development of efficient contrastive learning algorithms for graphs.

Thank you for your time and consideration!

The authors

---

### Meta-Review · Area_Chair_g9o7 · 2023-12-05

**Metareview:**

The paper proposes a new self-supervised learning  objective for graph neural network learning. The main contribution of this paper is cross-model contrastive learning, where positive samples are obtained through MLP and GNN representations from the central node and its neighbours, and negative pairs are from inter-model and intra-model representations. Experiments show the effectiveness of this work.

The main strength of this work is that its proposed cross-model contrastive learning can well construct positive and negative pairs for efficient learning.  However, one reviewer strongly reject this work because of the unclear definition of the scalability concept in the inference  and the missing inference time on the large graphs. Indeed, many other reviewers also pose many experimental issues which need the authors to add many extra new results.  This means this work may need big revision and is not ready to this conference. Consider the fact that many papers are around the borderline, including this one, and most reviewers have low intentions to accept (all positive scores are 6), we can only accept papers with high scores (>=6 typically), and cannot accept this work now. For improvement, the authors could well add the corresponding experiments and also introduce some key concepts more clear.

**Justification For Why Not Higher Score:**

1) one reviewer strongly reject this work because of the unclear definition of the scalability concept in the inference  and the missing inference time on the large graphs.

2) many other reviewers also pose many experimental issues which need the authors to add many extra new results.  This means this work may need big revision and is not ready to this conference.

3) most reviewers have low intentions to accept (all positive scores are 6), we can only accept papers with high scores (>=6 typically), and cannot accept this work now.

**Justification For Why Not Lower Score:**

N/A

---

### Decision · Program_Chairs · 2024-01-16

Reject